# SPATIO-TEMPORAL TWINS WITH A CACHE FOR MODELING LONG-TERM SYSTEM DYNAMICS

## ABSTRACT

This paper investigates the problem of modeling long-term dynamical systems, which are essential for comprehending fluid dynamics, astrophysics and earth science, *etc*. Recently, a variety of spatio-temporal forecasting approaches have emerged, typically leveraging complex architectures like Transformers to capture spatial and temporal correlations. However, these approaches often fail in long-term forecasting scenarios due to information loss during semantics exploration and iterative rollouts. Towards this end, we propose a new approach named Spatio-temporal Twins with a Cache (STAC) for long-term system dynamics modeling. STAC contains a discrete frequency-enhanced spatial module and an ODE-enhanced temporal module, which investigates spatio-temporal relationships from complementary perspectives. Then, we fuse the information between twin modules with channel attention at different granularities to generate informative feature maps. To enhance long-term prediction ability, we further introduce a cache-based recurrent propagator, which stores the previous feature maps in the cache memory during recursive updating. In addition, we optimize STAC with currently popular teacher forcing and adversarial learning techniques. We release a new flame flow field benchmark (`FIRE`) and conduct comprehensive validations across 14 benchmarks. Experimental results demonstrate that STAC shows superior performance in long-term spatio-temporal prediction and partial differential equation solving challenges. Our code avaiable at https://anonymous.4open.science/r/STAC-89A5.

## 1 INTRODUCTION

Dynamical systems (Thangamuthu et al., 2022) are pervasive in numerous disciplines including physics (Rao et al., 2023), biology (Li et al., 2022), and robotics (Allen et al., 2023). Modeling of these complex spatio-temporal system dynamics is a crucial problem in scientific research (Bueso et al., 2020; Gupta et al., 2022). A potential solution is to explicitly solve these fundamental equations using computational tools and simulation softwares, which could bring in enormous computational costs. With the rapid development of artificial intelligence, machine learning has recently demonstrated the capability of understanding complex dynamics from massive data.

In literature, numerous data-driven approaches are proposed to capture spatio-temporal dynamics (Wu et al., 2023; Raissi et al., 2019). Typically, these designs employ various neural architectures to model training data, achieving promising performance in various disciplines. For example, Transformer-based works (Ye & Bilodeau, 2023; Gao et al., 2022a; Liu et al., 2020) employ the attention mechanism to comprehend intricate spatio-temporal relationships. Graph neural networks (Sanchez-Gonzalez et al., 2020; Janny et al., 2023; Han et al., 2022b; Wang et al., 2023a) provide effective tools for learning from irregular meshes by employing the message passing mechanism.

In real-world scenarios, long-term forecasting of dynamical systems is a more challenging task with various applications (Sangrody et al., 2018; Nayak et al., 2023). Among them, fire fluid dynamics (Zhang et al., 2022) remains an underexplored yet crucial area, which usually requires long-term dynamics forecasting to ensure accurate planning. Towards this end, this work constructs a fire dynamics benchmark, where 3000kW tunnel fire is simulated using a simulator. However, long-term dynamics would make the spatio-temporal forecasting more challenging. Firstly, long-term dynamics usually accompany with highly complicated spatio-temporal signals. Existing approaches (Wu et al., 2023; Raissi et al., 2019; Gao et al., 2022a) usually learn signals in a discrete manner, which is incapable of capturing continuous spatio-temporal dynamics in nature. Secondly, long-term dynamics could contain temporal patterns from extreme local events in the past. How to capture and leverage

these historical information for effective system dynamics modeling remains an open problem. Thirdly, due to the difficulty of the task, existing predictors could generate unrealistic long-term trajectories with insufficient supervision and error accumulation (Zeng et al., 2023).

With the ability of accessing historical information, memory mechanism (Zhou et al., 2023) has a wide application in time-series forecasting (Chang et al., 2018), one-shot learning (Ni et al., 2019) and question answering (Sukhbaatar et al., 2015). Initially, memory networks use a memory component as a knowledge base to facilitate the inference phase of question-answering. The most frequently accessible data from the memory is stored in a cache, which operates as a buffer to speed up operations. Cache has been investigated for neural machine translation (Tu et al., 2018) and graph machine learning (Li & Chen, 2021) due to its efficacy. From this insight, it is highly anticipated to leverage a cache to facilitate modeling complex long-term system dynamics.

In this paper, we propose a novel framework named Spatio-temporal Twins with a Cache (STAC), which explores and reuses informative historical information for capturing long-term dynamics. In particular, STAC involves a frequency-enhanced spatial module and an ODE-enhanced temporal module to discover complicated spatio-temporal correlations from complementary perspectives. The frequency-enhanced spatial module combines the spatial and frequency domains, while the ODE-enhanced temporal module uses a neural ODE (Chen et al., 2018) to investigate continuous dynamics after fusing temporal and channel information. More importantly, we fuse the knowledge from twin modules with channel attention at different granularities to generate feature maps with rich semantics. To capture temporal patterns from historical data, we employ a cache-based recursive propagator that stores feature maps from prior timestamps. The current feature maps would be utilized by comparing them to the stored feature maps to perform recursive updating. These updated maps would be upsampled to produce the intended long-term trajectories. To refine the framework for accurate predictions, we not only employ a structural similarity metric to investigate the properties of states, but also generate additional trajectories without supervision, followed by adversarial learning for rational outcomes. Teacher forcing (Toomarian & Barhen, 1992; Gao et al., 2022b) with Mixup is also utilized to stabilize the learning process during the iterative updating. Extensive experiments demonstrate that the proposed STAC outperforms a range of methods across various research domains.

To summarize, this paper makes the following contributions: (1) *Fluid Benchmark.* We construct a new benchmark (`FIRE`) to model fire dynamics for long-term dynamic forecasting. (2) *New Perspective.* To the best of our knowledge, we are the first to incorporate cache memory concept into long-term system modeling. (3) *Novel Framework.* We propose a novel framework for modeling long-term system dynamics, which explores sufficient historical information with information fusion, trained with effective optimization strategies. (4) *Multifaceted Experiments.* Comprehensive experiments on various benchmark datasets demonstrate the effectiveness of the proposed STAC.

## 2 RELATED WORK

**Dynamical System Modeling.** Modeling dynamical systems (Thangamuthu et al., 2022; Wan et al., 2023; Chen et al., 2023) has been a long-standing problem in the machine learning community. A variety of deep learning methods have been devised to address this problem on both regular and non-regular grids. CNNs are commonly used to capture spatial relationships on regular grids (Fotiadis et al., 2020; Kim et al., 2019; Tompson et al., 2017; Rao et al., 2023), whereas GNNs can learn from irregular grids by adopting the message passing mechanism (Sanchez-Gonzalez et al., 2020; Janny et al., 2023; Han et al., 2022b). Further, a number of operators including the Fourier neural operator (Li et al., 2020) and their variants (Wen et al., 2022; Zhou et al., 2022) have demonstrated efficacy for PDE-based system modeling. However, these approaches often struggle to capture the long-term dynamics of dynamical systems, as they frequently encounter challenges like error accumulation and a lack of sufficient memory.

**Memory Network.** Memory networks have been extensively utilized to provide additional information for reading and writing (Weston et al., 2014; Shi et al., 2019; Xu et al., 2018). Early attempts of memory networks typically concentrate on question-answering (Bordes et al., 2015), accompanied by an attention mechanism for information retrieval. Recent developments in memory network include knowledge tracing (Zhang et al., 2017), graph neural networks (Xu et al., 2022), few-shot learning (Zhu & Yang, 2020), and medical image segmentation (Zhou et al., 2023). In this paper, cache memory is incorporated into iterative rollouts, which updates feature maps with enhanced long-term semantics for accurately modeling system dynamics.

# 3 NEW FLAME FLOW FIELD BENCHMARK FOR FLOW SYSTEM MODELING

Figure 1: Overall descriptions of this Benchmark.

**Data Collection and Processing.** In this paper, we focus on complex flow dynamics and long-term modeling issues in flow fields. We open a new *Flame flow field benchmark*, FIRE, to assess STAC ability to overcome the above challenge. Fire is crafted from the well-regarded industrial software, Fire Dynamics Simulator (FDS) (Wu et al., 2006) for simulations, to create a digital database that encompasses tunnel fire scenarios under various fire locations, scales, and ventilation conditions. Throughout the progression of the fire, we record temperature data at multiple points near the top of the tunnel, emulating the response of thermal sensors in real-world tunnels. Subsequently, we perform extensive post-processing on the sensor data along with thousands of temperature and smoke field images, culminating in a vast numerical database.

**Statistics of FIRE.** As shown in Figure 1, we simulate a road tunnel with dimensions of 50 m × 10 m × 5 m. The fire source is modeled as a propane gas fire with a maximum HRR of 20 $MW$ and increasing at $t^2$ . The fire source is modeled as a propane gas fire with dimensions of 4.6 m × 1.8 m × 2.4m. The maximum HRR is 20 $MW$ and increases at a rate of $t^2$. We study four scenarios in which the HRR reaches 20 $MW$ and stabilizes, requiring 2626s, 1307s, 653s, and 326s, respectively, to reach this state. The final fire simulation captures temperature and smoke patterns in the tunnel, including phenomena such as the fire plume and front, totaling 100GB.

# 4 THE PROPOSED APPROACH

**Problem Definition.** We aim to utilize a data-driven method to model the spatio-temporal evolution in dynamical systems. Given a system, $\boldsymbol{x} \in \Omega$ and $t \in \mathcal{T}$ denotes the spatial and time coordinates, $\boldsymbol{u}(x,t) \in \mathbb{R}^d$ is the state variable defined in the spatio-temporal domain $\Omega \times \mathcal{T}$. The system is could be governed by a set of PDEs or other complex rules. In this work, we study the regular mesh structures where $\boldsymbol{u}(x,t)$ is limited on standard discretized Cartesian grids. Given the states in the interval $[0, T^{obs}]$, we aim to predict the most probable future state in $[T^{obs} + 1, T]$ where $T$ is much larger than $T^{obs}$. Here, we utilize a tensor $\boldsymbol{u}^{(t)} \in \mathbb{R}^{C \times H \times W}$ to denote the state at the time step $t$.

## 4.1 FRAMEWORK OVERVIEW

This work investigates the problem of modeling long-term dynamical systems and proposes a novel approach named STAC to solve the problem. The high-level idea of STAC is to explore and reuse informative historical feature maps for capturing long-term dynamics. In particular, STAC introduces a frequency-enhanced spatial module and an ODE-enhanced temporal module to sufficiently discover spatio-temporal correlations from complementary perspectives. The extracted semantics would be fused at different granularities to produce informative feature maps. Then, STAC employs a cache-based recursive propagator that stores feature maps from prior timestamps, which would be compared with current feature maps for recursive updating. Finally, we utilize a comprehensive optimization framework including investigating a structural similarity metric and adversarial learning for reliable long-term predictions. An overview of our STAC can be found in Figure 2.

## 4.2 TWIN SPATIO-TEMPORAL ENCODER

Previous methods usually utilize Transformer-based architectures and Fourier neural operators to learn spatio-temporal relationships (Tran et al., 2023a; Pathak et al., 2022; Wang et al., 2023b). However, these methods usually take discrete neural architectures, which are difficult to capture continuous spatio-temporal dynamics in nature. Moreover, these discontinuous latent could fail to learn long-term complicated periodic signals, which results in inferior predictions. Towards this end,

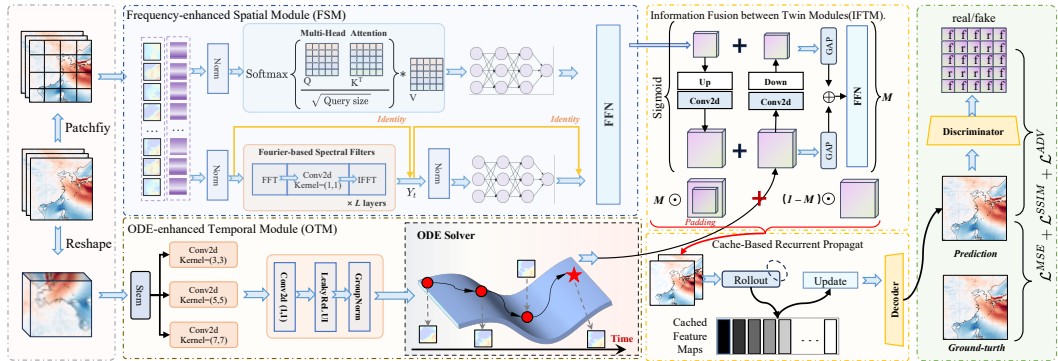

Figure 2: STAC overview. STAC contains a frequency-enhanced spatial module and an ODE-enhanced temporal module to learn spatio-temporal relationships. Then, STAC fuses information from twin modules at different granularities. For long-term dynamics, STAC employ a cache-based recursive propagator that compare current feature map with the stored ones. We utilize adversarial learning to enhance the reliability of long-term predictions.

besides a spatio-frequency discrete encoder, we introduce a ODE-based continuous module to learn continuous temporal dynamics as a complementary.

**Frequency-enhanced Spatial Module (FSM).** Fourier neural operator (Li et al., 2020) is an effective way to extract spectral information in Fourier space, which has shown both effectiveness and efficiency in dynamical system modeling. In short, we not only utilize a vision Transformer (Han et al., 2022a) to learn semantics in the spatial domain, but also introduce a Fourier neural operator to capture semantics in the frequency domain.

In particular, the input at every timestamp is denoted as $I^{in} \in \mathbb{R}^{C \times H \times W}$. We utilize CNNs to generate query, key and value vectors for a Transformer architecture, followed by a feed-forward network (FFN) to learn spatial correlations. Moreover, a 2D fast Fourier transform (FFT) is first leveraged to generate the frequency domain representation. Then, the complex-valued representation is then split into its real and imaginary parts and concatenated along the channel dimension. To enhance the integration of feature information, we utilize FFNs for token mixing across different channels, which allows for richer Fourier mode representations to emerge through greater fusion of channel-wise signals. Then, the mixed representations are then transformed back to the spatial domain using the 2D inverse Fourier transform to obtain the output of spectral filter layers. Finally, we concatenate the output of both Transformer and Fourier neural operator and then use a FFN as the final output. In formulation,

$$I^{out} = \text{FFN}([\text{Transformer}(I^{in}), \text{FNO}(I^{in})]), \quad (1)$$

where $\text{Transformer}(\cdot)$ and $\text{FNO}(\cdot)$ denote the Transform-based branch and frequency-based branch, respectively. $I^{out} \in \mathbb{R}^{l \times d}$, where $l = \hat{h} \cdot \hat{w}$ with the shape of output feature map $\hat{h} \times \hat{w}$.

**ODE-enhanced Temporal Module (OTM).** More importantly, we introduce a neural ODE module to capture continues dynamics in systems. Neural ODE is initially proposed as an approximation of ResNet (Chen et al., 2018), and has shown superior performance in time-series forecasting (Norcliffe et al., 2020; Bishnoi et al., 2022). To inject different levels of spatio-temporal relationships, we also pre-process the input into feature maps and then feed them into our neural ODE module with a multi-scale CNN, which can allow the capture of long-range correlations naturally with robustness

In particular, we introduce a stem module which first reshape the input tensor with $T^{obs}$ timestamps $\bar{I}^{in} \in \mathbb{R}^{T^{obs} \times C \times H \times W}$ into $\mathbb{R}^{T^{obs}C \times H \times W}$ and then utilize a CNN to generate feature maps $F_0 \in \mathbb{R}^{T^{obs}c \times h \times w}$ as follows:

$$F_0 = \text{CNN}(\text{Reshape}(\bar{I}^{in})). \quad (2)$$

However, STAC are in short of network depth to learn fine-grained and long-term dynamics. One solution is to stack a range of residual convolution blocks, which can be formulated as:

$$F_t = F_{t-1} + \text{Pad}(\text{MultiCNN}(F_{t-1})), \quad (3)$$

where $\mathrm{MultiCNN}(\cdot)$ is a multi-scale-CNN operator for diverse spatio-temporal semantics mining and $\mathrm{Pad}$ is to add zero-padding for residual connection. Although effective, this architecture has two limitations as follows. To begin, the discrete architecture is hard to capture continuous dynamics in natural systems. Worse yet, introducing a range of convolution blocks would bring a huge computational cost. To tackle this, we involve a neural ODE, which takes the $F_0$ as the initial state to learn continuous dynamics and update states as follow:

$$F_{T_t} = F_0 + \int_0^{T_t} \mathrm{Pad}(\mathrm{MultiCNN}\,(F_t))\mathrm{d}t, \tag{4}$$

where $T_t$ denotes the predefined terminate time with $F_{T_t} \in \mathbb{R}^{T^{obs} \times c \times h \times w}$. We can obtain $F_{T_t}$ from Eqn. 4 using an ODE solver (Chen et al., 2018).

**Information Fusion between Twin Modules (IFTM).** We introduce an effective module to fuse the information from twin modules. To begin with, our purpose is to capture global semantics to ensure the consistency and continuity in long-term predictions. However, the emphasis of global semantics would inevitably loss crucial details in local semantics. To balance the trade-off, we propose both fine-grainted fusion and coarse-grained fusion to tackle feature maps with different granularities.

We aggregate all $T^{obs}$ from the frequency-enhanced spatial module into $P^{\mathcal{S}} \in \mathbb{R}^{T^{obs} \times d \times \hat{h} \times \hat{w}}$, while the output of our ODE-enhanced temporal module is represented as $P^{\mathcal{T}} \in \mathbb{R}^{T^{obs} \times c \times h \times w}$. The fusion is primarily a blend of *fine-grainted fusion*, which upsamples $P^{\mathcal{S}}$ to match the spatial scale of $P^{\mathcal{T}}$, and *coarse-grained*, which downsamples $P^{\mathcal{T}}$ to match high-resolution features $P^{\mathcal{S}}$:

$$P_{i,j,k}^{up} = \mathrm{Upsample}(\mathrm{Conv}(P_{i,j,k}^{S})), \quad Q_{i,j,k}^{\mathcal{S}} = P_{i,j,k}^{up} + P_{i,j,k}^{\mathcal{T}}, \tag{5}$$

$$P_{i,j,k}^{down} = \mathrm{Downsample}(\mathrm{Conv}(P_{i,j,k}^{T})), \quad Q_{i,j,k}^{\mathcal{T}} = P_{i,j,k}^{down} + P_{i,j,k}^{\mathcal{S}}, \tag{6}$$

with $Q^{\mathcal{S}} \in \mathbb{R}^{T^{obs} \times K \times \hat{h} \times \hat{w}}$ and $Q^{\mathcal{T}} \in \mathbb{R}^{T^{obs} \times K \times h \times w}$. Given that channels in dynamic systems often capture diverse physical phenomena, we introduce a channel attention mechanism to aggregate diverse knowledge. We generate a weight matrix $M \in R^{T^{obs} \times K}$ as:

$$M = \sigma(\mathrm{FFN}(\mathrm{GlobalAvgPool}(Q^{\mathcal{S}}) \oplus \mathrm{GlobalAvgPool}(Q^{\mathcal{T}}))), \tag{7}$$

where $\oplus$ denotes the element-wise addition and $\mathrm{GlobalAvgPool}(\cdot)$ denotes the global average pooling operator. Then, the contributions from both modules are balanced to generate the final feature maps $X \in \mathbb{R}^{T^{obs} \times K \times h \times w}$:

$$X = M \odot \mathrm{Pad}(Q^{\mathcal{S}}) + (1 - M) \odot Q^{\mathcal{T}}, \tag{8}$$

where $\odot$ denotes the element-wise product. In this way, we can generate informative features maps with both coarse-grained and fine-grainted semantics.

## 4.3 Cache-based Recurrent Propagator

Previous methods usually output the short-term predictions at one time (Pathak et al., 2022; Wang et al., 2023b; Lee et al., 2021) and then utilize the rollout strategy by feeding the predicted trajectory into the model to make long-term prediction, which could perform poor due to the forgetting of past events. To tackle this, we propose a novel module based on the cache mechanism, which stores the previous representations in the memory updated with a first-in-first-out (FIFO) manner.

In detail, we first cut the whole $[T^{obs} + 1, T]$ into several pieces, i.e., $[t_0, t_1], [t_1, t_2], \cdots, [t_{M-1}, t_M]$. We denote the input, feature map and output for the interval $[t_{m-1}, t_m]$ as $I_m, \hat{I}_m$ and $X_m$, respectively. Rollout-based methods first utilize $[0, T^{obs}]$ to predict the trajectory in the interval $[t_0, t_1]$ and feed the prediction $\hat{I}_m$ back for $\hat{I}_{m+1}$ in a recursive manner. In virtue of last prediction, recurrent models take the feature maps $X_{m-1}$ and updated maps $Q_{m-1}$ at the last interval. The update rule in the hidden state can be written as:

$$A_m = f\,(Q_{m-1}, X_m)\,, Q_m = g(Q_{m-1}, A_m), \tag{9}$$

where $f(\cdot)$ and $g(\cdot)$ can be implemented with any RNN cell including Gated Recurrent Unit (GRU) (Dey & Salem, 2017) and Long Short-Term Memory (LSTM) (Graves & Graves, 2012). However, these updating rule could lose crucial long-term temporal dynamics. Towards this end,

we introduce cached feature maps. In particular, the cache $\mathcal{M}$ contains the most recent $R$ historical feature maps, i.e., $\{X_{m-R-1}, \cdots, X_{m-2}\}$, where $R$ denotes the cache size. The, the updating rule can be written as:

$$A_m = f(Q_{m-1}, X_m), W_m = g(Q_{m-1}, A_m), Q_m = \alpha W_m + (1-\alpha) \sum_{r=1}^{R} \phi(W_m, X_{m-R-2+r}), \quad (10)$$

where $\phi(\cdot)$ calculates the interaction between current features map and historical feature maps. $\alpha$ is a parameter to balance two parts. In our implementation, we would pre-allocate the memory for the cache module to keep the efficiency. Our cache-based propagator makes use of the cache memory to restore historical feature maps, which can capture long-term dynamics in the hidden space during recursive updating.

### 4.4 DECODER AND OPTIMIZATION

Finally, we introduce a decoder with unConvNormReLU blocks and LayerNorm (Tan et al., 2022), which transforms updated feature maps into predicted trajectories, i.e., $\hat{Y}_m = \text{Dec}(Q_m)$.

**Supervised Learning.** To begin, we minimize the vanilla MSE loss between the predictions and the target trajectories, i.e., $\mathcal{L}^{MSE} = \sum_{m=1}^{M} |Y_m - \hat{Y}_m|$. However, the MSE loss is sometimes too hard to be minimized. To tackle this, we introduce structural similarity metric (SSIM) (Zujovic et al., 2009) as a complementary, which calculates luminance, contrast and structure between the prediction. The formal formulation of the SSIM loss $\mathcal{L}^{SSIM}$ can be found in Appendix A.2.

**Teacher Forcing.** Moreover, we introduce the teacher forcing strategy (Toomarian & Barhen, 1992) with Mixup to stabilize the learning process between the recursive updating. In particular, we mix the prediction and ground truth before feeding them back to the model:

$$\tilde{I}_m := (1-\beta)I_m + \beta\hat{I}_m, \quad (11)$$

where $0 \leq \beta \leq 1$ is a dynamic parameter rising to 1 gradually during the optimization. Through this, we begin with short-term predictions with ground-truth middle input and end with full long-term predictions following the idea of curriculum learning (Bengio et al., 2009).

**Semi-supervised Adversarial Learning.** Another challenge for long-term prediction is that they could generate unrealistic trajectories. Toward this end, given that we have limited training data, we generate extra trajectories without supervision and utilize the idea of adversarial learning (Lowd & Meek, 2005) to make sure that our model can always generate rational results.

To begin, we introduce a discriminator $D(\cdot)$ conditional on the trajectories, which classifies real observations into positives and the generated observations into negatives. The whole adversarial learning objective can be formulated as:

$$\min_{\Theta} \max_{\Psi} \mathcal{L}^{ADV} = -\sum_{m=1}^{M} \log D(Y_m) - \sum_{m=1}^{M'} \log D(\hat{Y}_m), \quad (12)$$

where $\Theta$ and $\Psi$ denote the parameters of the discriminator and predictor, respectively. Note that $M' > M$ means that we introduce a range of predicted trajectories without ground truth, which helps increase the facticity of the predictor in a semi-supervised manner. We minimize $\mathcal{L}^{ADV}$ with respect to $\Theta$ for a well-trained discriminator. Further, we maximize $\mathcal{L}^{ADV}$ with respect to $\Psi$ for a more realistic predictor. The whole objective will summarize all three objectives as:

$$\mathcal{L} = \mathcal{L}^{MSE} + \mathcal{L}^{SSIM} + \mathcal{L}^{ADV}. \quad (13)$$

We summarize our algorithm in Appendix A.3.

## 5 EXPERIMENT

In this part, we undertake a thorough evaluation of the STAC's performance across benchmarks encompassing a broad spectrum of applications. These benchmarks encapsulates both simulated and real-world *physical dynamical systems*, which are extensively utilized in spatio-temporal modeling. We tested fourteen physical dynamical systems to evaluate the effectivenesses of our proposal. For ease of understanding, we present the five most challenging datasets in the main text, while the complete results are provided in the Appendix B.

- **`FIRE dataset`** is created using the Fire Dynamics Simulator (FDS) to simulate a 3000kW tunnel fire. We **release this benchmark** by collecting velocity and temperature data.
- **`Turbulence dataset`** (Khojasteh et al., 2022) analyzes flow patterns downstream of a cylinder at a Reynolds number of 3900 using Lagrangian and Eulerian methods.
- **`ERA5 dataset`** (Rasp et al., 2023) is a high-resolution global atmospheric reanalysis dataset used to assess the model's efficiency in modeling highly nonlinear and chaotic dynamical systems for climate research and weather predictions.
- **`SEVIR dataset`** (Veillette et al., 2020)represents a comprehensive storm event imagery collection, designed specifically for deep learning endeavors in both radar and satellite meteorology.
- **`KTH dataset`** (Schuldt et al., 2004) captures human kinematics and stands as a benchmark in the video prediction realm, it is used to gauge the model's performance in the visual domain.

**Baselines.** Our STAC would be compared with a range of state-of-the-art approaches including FNO (Li et al., 2020), F-FNO (Tran et al., 2023b), E3D-LSTM (Wang et al., 2018), MIM (Wang et al., 2019), PredRNN-v2 (Wang et al., 2023b), SimVP-V2 (Tan et al., 2022), Earthformer (Gao et al., 2022b), and FourcastNet (Kurth et al., 2023).

**Research Questions (RQ).** In this paper, we endeavor to answer the following research questions through extensive experimentation: (I) How effective is STAC and can it adeptly address long-term modeling challenges? (II) Can STAC effectively deal with complex system dynamics modeling problems? (III) Does STAC have the ability to sense extreme local events? (IV) Does STAC have the potential to become the backbone in the realm of video prediction?

Table 1: The average results from five runs of STAC were compared with the baseline. In this study, the selected evaluation metrics are MSE (Mean Squared Error) and MAE (Mean Absolute Error). For both metrics, lower values are preferable. The best results are highlighted in **bold**.

| Backbone | STAC | | FNO (2020) | | F-FNO (2023a) | | E3D-LSTM (2018) | | MIM (2019) | | PredRNN-V2 (2023b) | | SimVP-V2 (2022) | |
|---|---|---|---|---|---|---|---|---|---|---|---|---|---|---|
| | MSE | MAE | MSE | MAE | MSE | MAE | MSE | MAE | MSE | MAE | MSE | MAE | MSE | MAE |
| *Learning rate = 0.01; Optimizer = Adam; Attention head in FSM = 4; Hidden layer dimension = 64.* | | | | | | | | | | | | | | |
| Turbulence | **0.5123** | **0.5345** | 0.6567 | 0.7789 | 0.8124 | 0.9876 | 1.1234 | 1.4567 | 0.8321 | 0.9472 | 1.0163 | 1.0987 | 1.2765 | 1.4321 |
| ERA5 | **1.9865** | **1.7791** | 2.8534 | 2.2983 | 8.9853 | 7.34317 | 3.0952 | 2.9854 | 3.3567 | 3.2236 | 2.2731 | 2.6453 | 3.0843 | 3.0743 |
| SEVIR | **1.9731** | **1.4054** | 3.0833 | 1.8831 | 10.9831 | 5.4432 | 4.1702 | 2.5563 | 3.9842 | 2.0012 | 3.9014 | 1.9757 | 2.9371 | 1.7743 |
| Fire | **0.5493** | **0.7217** | 0.9985 | 1.0432 | 2.7412 | 1.6557 | 1.0921 | 0.8731 | 1.8743 | 1.5324 | 0.7789 | 0.6863 | 1.7743 | 1.0321 |
| KTH | **28.8321** | **24.2216** | 33.1983 | 29.7421 | 31.8741 | 29.8753 | 86.1743 | 85.5563 | 56.5942 | 54.8426 | 51.1512 | 50.6457 | 40.8421 | 43.2931 |

## 5.1 MAIN RESULTS TO ANSWER (**RQ I**)

To answer RQ I, we conduct the in-depth comparative analyses involving STAC and various SOTA models, including PDE modeling, ST prediction, and the application of certain domain-specific methods to dynamical system modeling tasks. The comparison results appear in the Tab. 1. Further, we choose the `Turbulence` as a representative of complex systems for the experiment due to it irregularity and unpredictability. Figure 3 shows the comparison with FNO and the trend of MSE with the predicted step size. Based on results, we can summarize the following **Obs**ervations:

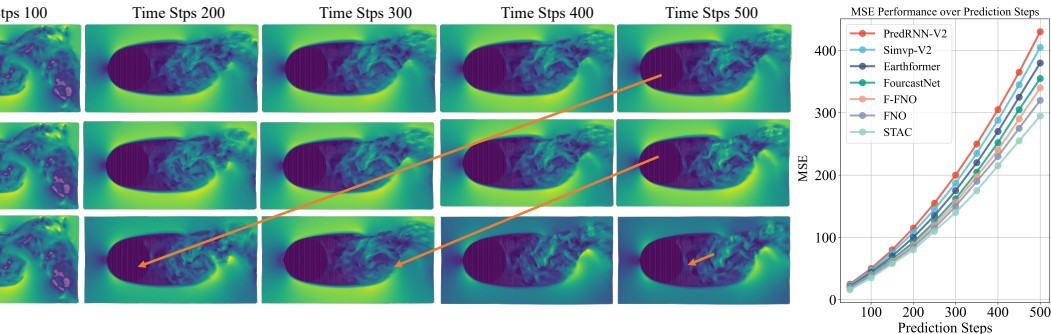

Figure 3: Visualization of the Turbulence dataset spans various timestamps. We represent the composite speed, computed as speed $= \sqrt{u^2 + v^2}$, within the fluid field of the STAC model, alongside the display of the ground truth for thorough comparison and evaluation.

**Obs.1.** STAC consistently demonstrates outstanding performance across various datasets in terms of MSE and MAE metrics. Notably, on the ERA5 dataset, compared to the existing best method,

PredRNN-V2, STAC achieved nearly 14.4% decline in MSE and an impressive 29.1% decline in MAE. Whether on Turbulence, SEVIR, or other datasets, STAC consistently showcases its superior predictive capabilities, underscoring its efficacy.

**Obs.2.** In Figure 3, we employ a fixed-window strategy where models evaluate within a forecast range of 100, 200, 300, 400, and 500 time steps, reflecting long-term forecasting challenge. It is observed that STAC achieves the lowest MSE among the seven models. Furthermore, as the prediction term increases, STAC consistently exhibits a stronger performance advantage. Even when compared to the currently best FNO, STAC maintains a lower MSE, indirectly demonstrating STAC's superiority in long-term prediction challenges. More detailed results are presented in the Appendix E.

## 5.2 CAN STAC EFFECTIVELY MODEL COMPLEX SYSTEM DYNAMICS? (**RQ II**)

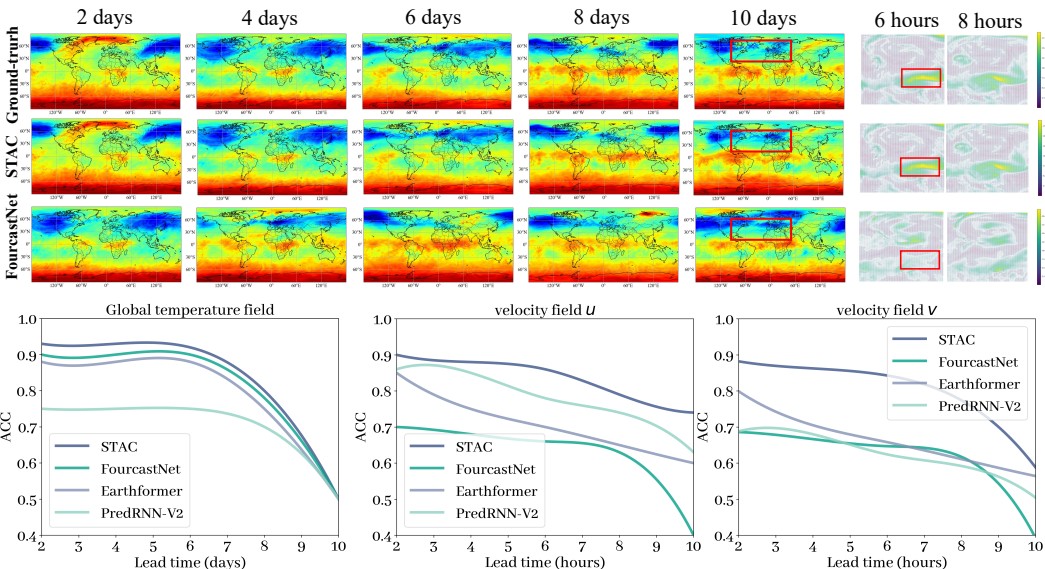

Figure 4: Visualization of the ERA5. On the left is the global temperature field, while on the right lies the localized velocity field. The line chart depicts the variation in ACC (Anomaly Correlation Coefficien) for both STAC and the baseline models as the prediction time step lengthens.

To answer RQ II, we chose ERA5 as our primary dataset for analysis. Its rich detail and diverse data dimensions make it an ideal choice for illustrating and investigating complex climate and atmospheric dynamics processes. We select representative SOTA models in the field of earth science as comprehensive assessment, *i.e.,* FourcastNet, Earthformer, and advanced ST sequence forecasting models like PredRNN-V2. The relevant visualization results and experimental data line graphs are shown in the Figure 4, in which we can made the following observations and findings:

**Obs.3.** With the forecast time step increases, FourcastNet shows significant oversmoothing, especially over the 10-day forecast. In the same circumstance, STAC can preserves local details via **IFTM**. FourcastNet performs poorly in short-term prediction of complex systems. Inversely, STAC is better at modeling dynamic systems from both global and local perspective.

**Obs.4.** Both in meteorological models and advanced ST predictions settings, STAC consistently excels in the ACC metric. As prediction steps increased, STAC's performance degradation is the slowest. Notably, in predicting the velocity field V, STAC's decline becomes significant after 8 hours, yet it still outperforms the second-best model, earthformer, by approximately 11.1%.

## 5.3 DOES STAC HAVE THE ABILITY TO SENSE EXTREME LOCAL EVENTS? (**RQ III**)

To answer RQ III, we select the SEVIR and the FLAME FLOW FIELD dataset to focus on the analysis. We focus on whether SEVIR's storm details are successfully captured, and examine the rapid spread of flames, temperature fluctuations, and the movement of smoke in the tunnel environment. As shown in the Figure 5, we make the following observations:

**Obs.5.** When we zoom in on the predictive details of SEVIR, it's clear that STAC presents a sharper rendition compared to Earthformer's relative blurriness. Notably, at 240s, STAC precisely forecasts

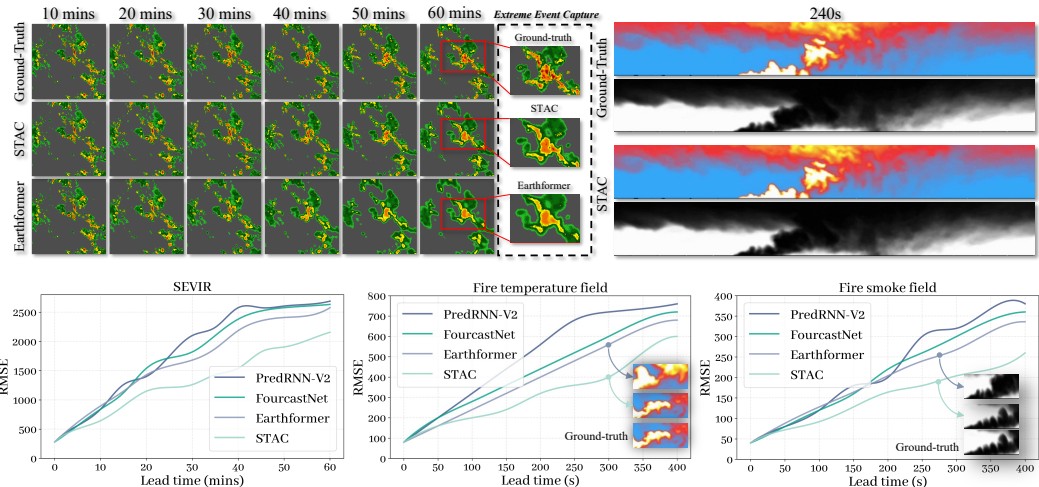

Figure 5: Visualization of the SEVIR and FLAME FLOW FIELD datasets in the first row, followed by line plots illustrating the variation of RMSE with prediction time in the second row.

the overarching trend of the tunnel fire's temperature and smoke dynamics. In the corresponding line graph, by the 300s mark, STAC effectively counteracts the noticeable lag exhibited by Earthformer.

## 5.4 CAN STAC BECOME THE BACKBONE OF VIDEO PREDICTION? (**RQ IV**)

Input length = 10

output length = 10

| Datasets (KTH) | | Output Sequence | | |
|---|---|---|---|---|
| Backbone | Input Seq (10) | → 10 | → 15 | → 20 |
| PredRNN-V2 | SSIM | 88.67 | 86.53 | 84.22 |
| | PSNR | 29.98 | 28.47 | 25.23 |
| SimVP-V2 | SSIM | 92.43 | 91.33 | 89.89 |
| | PSNR | 36.31 | 35.54 | 33.65 |
| STAC | SSIM | 93.72 | 92.83 | 90.07 |
| | PSNR | 37.29 | 36.87 | 34.32 |

Figure 6: The KTH performance visualizations: the second row is the ground-truth, the third is the STAC prediction results, the fourth row is the SimVP-V2 prediction results, and the fifth row is the PredRNN-V2 prediction results.

Table 2: Performance comparison of various models on the KTH dataset, evaluated using SSIM and PSNR metrics.

In the last, we consider whether STAC serves as a universal framework for video prediction. We study the KTH video dataset, which is an action recognition library that contains movements of various subjects in different contexts. As the Figure 6 shows, we find the following:

**Obs.6.** STAC model significantly outperforms both PredRNN-V2 and SimVP-V2. Specifically, for 20-frame predictions, compared to PredRNN-V2, STAC's SSIM increased by 6% and its PSNR rose by 9 units. In contrast to SimVP-V2, SSIM and PSNR respectively improved by 0.2% and 7 units. This highlights STAC's advantage in key metrics, showcasing its exceptional predictive accuracy and image quality. We summarize more ablations in Appendix F for integrity of our studies.

## 6 CONCLUSION

This paper studies the problem of modeling long-term dynamical systems and proposes a new paradigm to solve this problem. Our STAC consists of a discrete frequency-enhanced spatial module and an ODE-enhanced temporal module that investigates spatio-temporal relationships from contrasting viewpoints. Then, we fuse information between twin modules with different granularities to generate informative feature maps. To improve the capacity to make long-term predictions, we introduce a cache-based recurrent propagator that stores the prior feature maps in the cache memory during recursive updating. We release a new benchmark FIRE and extensive experiments on various benchmarks validate the superiority of STAC. In future works, we would extend STAC to more scenarios such as mesh-based physical simulations and pedestrian trajectory prediction.

ETHICS STATEMENT

We acknowledge that all co-authors of this work have read and committed to adhering to the ICLR Code of Ethics.

REPRODUCIBILITY STATEMENT

To increase reproducibility, we have provided all the details of the proposed STAC in Appendix A. Our code is available at https://anonymous.4open.science/r/STAC-89A5 anonymously.

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

# A MORE METHODOLOGY DETAILS

## A.1 DECODER

The dimension of the feature map is $H/16$ and $W/16$. After going through the Decoder operation, the output dimension becomes $H$ and $W$, which is the target dimension. The detailed process is as follows:

$$X_0 \in \mathbb{R}^{B \times \text{embed\_dim} \times \frac{H}{16} \times \frac{W}{16}} \tag{14}$$

Here, $X_0$ is the tensor input to this module.

$$X_1 = \text{Tanh}(\text{ConvTranspose2d}(X_0)) \in \mathbb{R}^{B \times \text{out\_channels} \times \frac{H}{8} \times \frac{W}{8}} \tag{15}$$

In this step, we use a transposed convolution layer followed by a Tanh activation function to obtain the tensor $X_1$.

$$X_2 = \text{Tanh}(\text{ConvTranspose2d}(X_1)) \in \mathbb{R}^{B \times \text{out\_channels} \times \frac{H}{4} \times \frac{W}{4}} \tag{16}$$

Next, we again use a transposed convolution layer and Tanh activation function to get the tensor $X_2$.

$$X_3 = \text{ConvTranspose2d}(X_2) \in \mathbb{R}^{B \times \text{out\_channels} \times H \times W} \tag{17}$$

Finally, we apply another transposed convolution layer to get the output tensor $X_3$ with dimensions $H$ and $W$.

## A.2 LOSS FUNCTION DETAILS

Our loss function is as follows, inside the main text, we have described $\mathcal{L}^{MSE}$ and $\mathcal{L}^{ADV}$ in detail, next, we describe $\mathcal{L}^{SSIM}$ in detail.

$$\mathcal{L} = \mathcal{L}^{MSE} + \mathcal{L}^{SSIM} + \mathcal{L}^{ADV}. \tag{18}$$

SSIM (Structural Similarity Index) is a metric used to assess the structural similarity of two images. It is proposed to better reflect the human eye's subjective perception of image quality, and provides a more intuitive and accurate assessment of image quality than the traditional mean square error (MSE) or peak signal-to-noise ratio (PSNR).

Specifically, two four-dimensional tensors of dimension $[T \times C \times H \times W]$ are given: predicted data $P$ and real labeled data $G$, where T stands for the time dimension or batch size, C is the number of channels, and H and W are the height and width of the tensor, respectively. To compute the SSIM loss, a window of fixed size $w$ (e.g., a Gaussian window of 11x11) and two constants for stabilizing the denominator $c_1$ and $c_2$ are first chosen.

For each sample $t$ and each channel $c$, we define the following calculations:

1. mean value:

$$\mu_{P_{t,c}} = w \cdot P_{t,c} \tag{19}$$

$$\mu_{G_{t,c}} = w \cdot G_{t,c} \tag{20}$$

2. variance:

$$\sigma^2_{P_{t,c}} = w \cdot P^2_{t,c} - \mu^2_{P_{t,c}} \tag{21}$$

$$\sigma^2_{G_{t,c}} = w \cdot G^2_{t,c} - \mu^2_{G_{t,c}} \tag{22}$$

3. covariance:

$$\sigma_{P_{t,c}G_{t,c}} = w \cdot P_{t,c} \cdot G_{t,c} - \mu_{P_{t,c}} \cdot \mu_{G_{t,c}} \tag{23}$$

Then, the SSIM value for each position is calculated using the SSIM formula:

$$\text{SSIM}_{t,c} = \frac{(2\mu_{P_{t,c}}\mu_{G_{t,c}} + c_1)(2\sigma_{P_{t,c}G_{t,c}} + c_2)}{(\mu^2_{P_{t,c}} + \mu^2_{G_{t,c}} + c_1)(\,sigma^2_{P_{t,c}} + \sigma^2_{G_{t,c}} + c_2)} \tag{24}$$

Average over all positions to obtain the SSIM value for that channel. Finally, average the SSIM values over all samples and channels and subtract this value from 1 to get the SSIM loss:

$$\mathcal{L}^{SSIM} = 1 - \text{mean}(\text{SSIM}_{t,c}) \tag{25}$$

For efficiency, the computation of mean, variance and covariance can be realized by convolution operation.

### A.3 ALGORITHMIC PROCESS

The proposed algorithm, named STAC, offers a novel approach to model the spatio-temporal evolution in dynamical systems. At its core, the algorithm integrates a twin spatio-temporal encoder, which captures both spatial and frequency domain semantics, and a cache-based recurrent propagator, which leverages historical data to enhance long-term dynamics prediction. The encoder consists of a Frequency-enhanced Spatial Module (FSM) and an ODE-enhanced Temporal Module (OTM), which are then fused together. The recurrent propagator utilizes a cache mechanism to store and update previous representations. Finally, a decoder transforms the updated feature maps into predicted trajectories, which are then optimized using a combination of loss functions. The algorithm aims to provide accurate and reliable long-term predictions for dynamical systems.

---

**Algorithm 1** The STAC Approach

---

0: **function** METHOD(Input: system states in interval $[0, T^{obs}]$) {Twin Spatio-temporal Encoder}
0:     $I^{in} \leftarrow$ Input
0:     $I^{out} \leftarrow$ FSM($I^{in}$) {Frequency-enhanced Spatial Module}
0:     $F_0 \leftarrow$ OTM($I^{in}$) {ODE-enhanced Temporal Module}
0:     $X \leftarrow$ IFTM($I^{out}, F_0$) {Information Fusion between Twin Modules}
    {Cache-based Recurrent Propagator}
0:     Initialize cache $\mathcal{M}$ with size $R$
0:     **for** $m = 1$ to $M$ **do**
0:         $Q_m \leftarrow$ UpdateCache($Q_{m-1}, X_m, \mathcal{M}$)
0:         Add $X_{m-1}$ to cache $\mathcal{M}$
0:     **end for**
    {Decoder and Optimization}
0:     $Y_{\text{hat}} \leftarrow$ Decode($Q_m$)
0:     Loss $\leftarrow$ CalculateLoss($Y_{\text{hat}}$, GroundTruth)
0:     UpdateModelParameters(Loss)
0:     **return** $Y_{\text{hat}}$
0: **end function**=0

---

## B    DETAILED DESCRIPTION OF BENCHMARKS

We summarize the benchmark configurations in Tab. Here are the details of the dataset.

### B.1    TURBULENCE DATASET

This dataset (Khojasteh et al., 2022) contains Eulerian velocity fields and pressure fields. An open-source direct numerical simulation (DNS) flow solver named Incompact3d was used to compute the Eulerian fields around the cylinder. Following the original thesis setup, highly resolved direct numerical simulations (DNS) of the flow over a smooth cylinder at a subcritical Reynolds number of 3900 (based on the diameter D of the cylinder and the diameter D of the freestream velocity) were performed to generate the data. Double-precision Eulerian and Lagrangian fields were collected for both subdomains as shown in Fig. 7. Due to online cloud storage limitations, every 10 DNS time steps

were saved every 10 DNS time steps (saving each time step would require about 30 TB of storage space per vortex shed). The 1000 snapshots are also used for smaller subdomains with dimensions of 4D x 2D x 2D (i.e., per DNS time step). Subdomain 2 is suitable for studies that require the highest possible temporal resolution. Detailed information on the two subdomains can be found in Table 2. An Eulerian snapshot of the current tail stream is shown in Fig. 2. For both subdomains, Lagrangian trajectories are provided for about 200000 synthetic particles. Three main categories are provided in the data repository: subdomain 1, subdomain 2, and software. Snapshots are in text format (.txt) and are collected in compressed files (.zip). There are no special requirements for reading and opening the data. Euler 3D snapshots are saved in vector format. Therefore, they need to be extracted within three internal loops in the xyz direction.

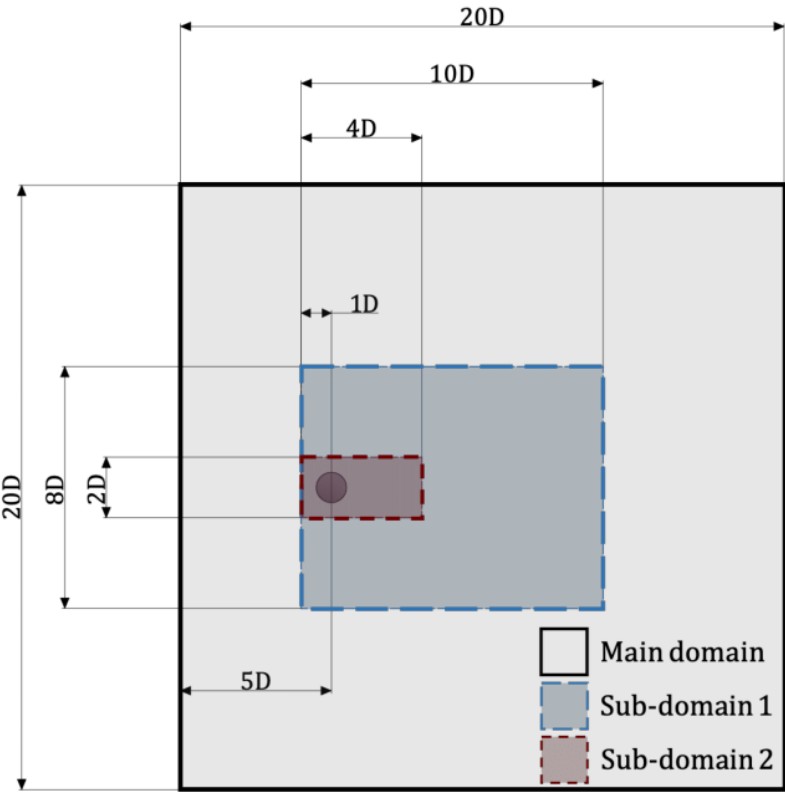

Figure 7: The flow around a smooth cylinder at a subcritical Reynolds number of 3900, with dimensions of two computational subdomains. (Khojasteh et al., 2022)

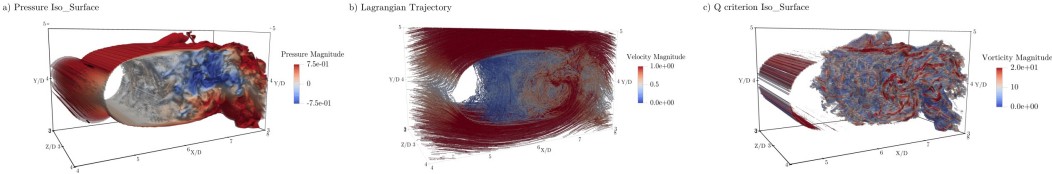

Figure 8: Snapshot Overview of Sub-domain 2: (a) Pressure iso-surface highlighted by the intensity of the pressure. (b) Lagrangian trajectories of 20,000 particles, visualized after 1,000 DNS time steps, color-coded by velocity magnitude. (c) Q-criterion representation depicting Eulerian flow structures, color-graded by the magnitude of vorticity.

## B.2 ERA5 DATASET

ERA5 is the latest global reanalysis product released by the European Centre for Medium-Range Weather Forecasts (ECMWF). It provides researchers and meteorologists with high-resolution meteorological and climatic data from 1979 to the present. The spatial resolution of ERA5 data is 31 kilometers, with a temporal resolution of hourly, representing a significant improvement over previous reanalysis products. It encompasses observations from the atmosphere, land, and oceans, offering invaluable data resources for global climate change, weather forecasting, and other related studies. The data quality and accuracy of ERA5 have been widely recognized by researchers, making it an essential tool in climate research and meteorological forecasting.

We have selected global temperature field data and local velocity field data as our training set. The former has a resolution of 1440x720. In our experiments, we downsampled it to a quarter of its original size, i.e., 360x180. The velocity field has a resolution of 64x64. For the temperature data, we employed autoregressive training, using 5 days of data as input and 10 days of data as output. For the velocity field, we chose 8 hours of data as input and another 8 hours of data as output.

## B.3 SEVIR DATASET

The Storm EVent ImagRy (SEVIR) dataset presents a meticulously curated set of spatiotemporally synchronized images, capturing meteorological phenomena via the GOES-16 geostationary satellite and the NEXRAD weather radar system. Encompassing in excess of 10,000 distinct weather events, each individual event in this collection showcases an image sequence persisting for a duration of 4 hours, spanning a geographical expanse of 384 km x 384 km. Delving into this dataset can expedite advancements in the realms of weather sensing, hazard avoidance, near-term forecasting, and other pertinent meteorological applications.

We follow the same setup as Earthformer, using 13 frames as input and 12 frames as output.

## B.4 FLAME FLOW FIELD DATASET

In this study, we select a typical highway tunnel for simulation, with dimensions of 50 meters in length, 10 meters in width, and 5 meters in height. The fire source has dimensions of 4.6 meters in length, 1.8 meters in width, and 2.4 meters in height. The top surface of the truck is set as a "burner" type. To simulate a realistic scenario, the maximum heat release rate (HRR) of the fire source is set at $20\ MW$, a value recommended by the standard for the maximum HRR of tunnel fires in the event of a truck fire. The fire source is modeled as a propane gas fire, with its HRR growing at a $t^2$ rate. Four operating conditions are designed, In all four scenarios, the power of the fire source is consistently $20\ MW$. In the first scenario, the fire source growth factor is 0.0029 $kW/s^2$, with the time to reach steady state being 2626 seconds and another steady state time being 2700 seconds. In the second scenario, the fire source growth factor is 0.0117 $kW/s^2$, with the times to reach steady state being 1307 seconds and 1400 seconds, respectively. In the third scenario, the fire source growth factor is 0.0469 $kW/s^2$, with the steady state times being 653 seconds and 700 seconds. Lastly, in the fourth scenario, the fire source growth factor is 0.1876 $kW/s^2$, with the times to reach steady state being 326 seconds and 400 seconds. The choice of actual tunnel dimensions, fire source size, and HRR values ensures the validity and relevance of the simulation results, providing a solid foundation for the proposed artificial intelligence fire prediction method.

In this study, the input dimensions are set at [10,2,80,480], while the output dimensions are [90,2,80,480]. Here, the input duration of 10 seconds represents the observation time, and the value of 2 corresponds to the temperature field and smoke field, both of which have a resolution of 80x480. The output duration of 90 seconds is used for extended time-range predictions. To achieve this long-term forecasting, we employ a rollout strategy. Moreover, the caching mechanism introduced in this paper plays a pivotal role in enhancing the accuracy and efficiency of long-term predictions.

## B.5 KTH DATASET

The KTH dataset stands as a benchmark in the domain of human activity recognition, stemming from the esteemed KTH Royal Institute of Technology in Sweden. This collection distinctly captures six

human activities: walking, jogging, running, boxing, hand waving, and hand clapping. Across diverse scenarios—ranging from outdoor settings (s1, s4) and scaled outdoor variations (s2, s3) to indoor environments (s5, s6)—25 participants, donning varied attire, repetitively perform these actions. Each video in this dataset is recorded at a clarity of 128x128 pixel resolution and maintains a consistent frame rate of 25 frames per second.

### B.6 DYNAMIC SYSTEM DATASETS RECORDED BY VIDEO.

We have provided 9 datasets of dynamic system, recorded in the form of videos. Both the input and output dimensions are [10,3,128,128], indicating an input length of 10 time steps and an output length of 10 time steps. Since they are recorded as videos, there are 3 channels, representing RGB, with a resolution of 128x128 for each image.

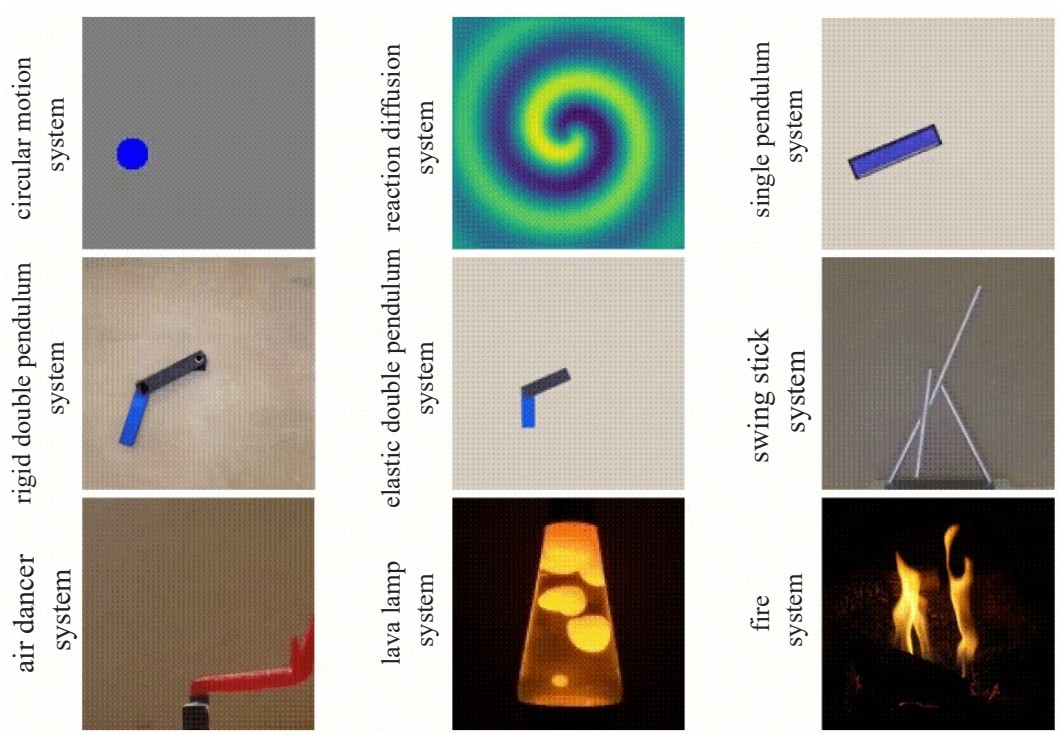

Figure 9: A case study of a dataset of dynamical systems recorded by video

**1. Circular motion system. (CMS)** An object moves uniformly along a fixed radius. Formula as follows:

$$v = r\omega, \tag{26}$$

$$a = r\omega^2, \tag{27}$$

where $v$ is the linear speed, $r$ is the radius of the circle, $\omega$ is the angular speed, and $a$ is the centripetal acceleration.

**2. Reaction diffusion system. (RDS)** A system describing how the concentration of substances changes over time due to reactions and diffusion. Formula as follows:

$$\frac{\partial u}{\partial t} = D_u \nabla^2 u - uv^2 + F(1 - u), \tag{28}$$

$$\frac{\partial v}{\partial t} = D_v \nabla^2 v + uv^2 - (F + k)v, \tag{29}$$

where $u$ and $v$ are concentrations, $D_u$ and $D_v$ are their diffusion coefficients, and $F$ and $k$ are system parameters.

**3. Single pendulum system. (SPS)**   A point mass hung from a fixed point swings due to gravity. Formula as follows:

$$\frac{d^2\theta}{dt^2} = -\frac{g}{l}\sin(\theta), \tag{30}$$

where $\theta$ is the pendulum angle, $l$ is the length of the pendulum, and $g$ is gravitational acceleration.

**4. Rigid double pendulum system. (RDPS)**   A complex pendulum system consisting of two pendulums, with one pendulum attached to the end of another. The equations of a double pendulum are usually relatively complex and involve multiple variables. But the basic idea is to use the Lagrange equation. For a simplified description:

$$\frac{d^2\theta_1}{dt^2} = \frac{g(\sin\theta_2\cos(\theta_1 - \theta_2) - \sin\theta_1) - (\ell_2\ddot{\theta}_2 + \ell_1\dot{\theta}_1^2\sin(\theta_1 - \theta_2))\cos(\theta_1 - \theta_2)}{\ell_1(\cos^2(\theta_1 - \theta_2) - 1)}, \tag{31}$$

$$\frac{d^2\theta_2}{dt^2} = \frac{g\sin\theta_1\cos(\theta_1 - \theta_2) - \ell_1\dot{\theta}_1^2\sin(\theta_1 - \theta_2) - g\sin\theta_2}{\ell_2(\cos^2(\theta_1 - \theta_2) - 1)}, \tag{32}$$

**5. Elastic double pendulum system. (EDPS)**   Similar to the double pendulum, but the connecting component between the pendulums is elastic. The basic mathematical description of an elastic pendulum involves Hooke's law of springs and the motion of a pendulum. A simplified description is:

$$\frac{d^2x}{dt^2} = -kx/m - g\sin\theta , \tag{33}$$

$$\frac{d^2\theta}{dt^2} = -g/x\cos\theta , \tag{34}$$

where $x$ is the displacement from the equilibrium position, $k$ is the spring constant, and $m$ is the mass.

**6. Swing stick system. (SSS)**   A long stick with a fixed endpoint that swings under the influence of gravity and other possible external forces. The pendulum system is equivalent to a long pendulum. The basic description is similar to a simple pendulum, but requires consideration of the mass distribution and length of the rod. The simplest description is:

$$\frac{d^2\theta}{dt^2} = -\frac{3g}{2L}\sin\theta, \tag{35}$$

where $L$ is half the length of the rod.

**7. Air Dancer System. (ADS)**   For the air dancer, it is crucial to consider the influence of the gas flow. This can be described by the incompressible Navier-Stokes equation:

$$\frac{\partial \mathbf{u}}{\partial t} + (\mathbf{u} \cdot \nabla)\mathbf{u} = -\frac{1}{\rho}\nabla p + \nu\nabla^2\mathbf{u}, \tag{36}$$

$$\nabla \cdot \mathbf{u} = 0, \tag{37}$$

where $\mathbf{u}$ is the velocity, $p$ is the pressure, $\rho$ is the density, and $\nu$ is the kinematic viscosity.

**8. Lava Lamp System. (LLS)**   At the heart of the lava lamp is the fluid flow caused by density changes due to temperature. This can be described using the Navier-Stokes equation and the Boussinesq approximation:

$$\frac{\partial \mathbf{u}}{\partial t} + (\mathbf{u} \cdot \nabla)\mathbf{u} = -\frac{1}{\rho_0}\nabla p + \nu\nabla^2\mathbf{u} - \alpha(T - T_0)\mathbf{g}, \tag{38}$$

$$\nabla \cdot \mathbf{u} = 0, \tag{39}$$

$$\frac{\partial T}{\partial t} + \mathbf{u} \cdot \nabla T = \kappa\nabla^2 T, \tag{40}$$

where $T$ is the temperature, $\alpha$ is the thermal expansion coefficient, $\kappa$ is the thermal diffusivity, and $\mathbf{g}$ is the acceleration due to gravity.

**9. Fire System. (FS)**   The fire system involves chemical reactions, heat transfer, and fluid dynamics. A common description is:

$$\frac{\partial \mathbf{u}}{\partial t} + (\mathbf{u} \cdot \nabla)\mathbf{u} = -\frac{1}{\rho}\nabla p + \nu\nabla^2\mathbf{u} + \text{source terms due to combustion}, \tag{41}$$

$$\nabla \cdot \mathbf{u} = 0, \tag{42}$$

$$\frac{\partial T}{\partial t} + \mathbf{u} \cdot \nabla T = \kappa\nabla^2 T + \text{source terms due to combustion}, \tag{43}$$

$$\frac{\partial Y_i}{\partial t} + \mathbf{u} \cdot \nabla Y_i = D\nabla^2 Y_i + \text{reaction rate of species } i, \tag{44}$$

where $Y_i$ is the mass fraction of the $i$-th chemical species, and $D$ is the diffusion coefficient.

## C   EXPERIMENTAL DETAILS

### C.1   EVALUATION METRICS

We utilize the following metrics:

Mean Squared Error (MSE) Given the predicted data dimension $Y_{\text{pred}} \in \mathbb{R}^{T \times C \times H \times W}$ and the label data dimension $Y_{\text{label}} \in \mathbb{R}^{T \times C \times H \times W}$, the MSE is computed as:

$$\text{MSE}(Y_{\text{pred}}, Y_{\text{label}}) = \frac{1}{T \times C \times H \times W}\sum_{t=1}^{T}\sum_{c=1}^{C}\sum_{h=1}^{H}\sum_{w=1}^{W}(Y_{\text{pred}}^{tchw} - Y_{\text{label}}^{tchw})^2 \tag{45}$$

Mean Absolute Error (MAE) The MAE is given by:

$$\text{MAE}(Y_{\text{pred}}, Y_{\text{label}}) = \frac{1}{T \times C \times H \times W}\sum_{t=1}^{T}\sum_{c=1}^{C}\sum_{h=1}^{H}\sum_{w=1}^{W}|Y_{\text{pred}}^{tchw} - Y_{\text{label}}^{tchw}| \tag{46}$$

Anomaly Correlation Coefficient (ACC) The ACC, often used in meteorology, is defined as:

$$\text{ACC} = \frac{\sum(Y_{\text{pred}} - \bar{Y_{\text{pred}}})(Y_{\text{label}} - \bar{Y_{\text{label}}})}{\sqrt{\sum(Y_{\text{pred}} - \bar{Y_{\text{pred}}})^2 \sum(Y_{\text{label}} - \bar{Y_{\text{label}}})^2}} \tag{47}$$

where $\bar{Y_{\text{pred}}}$ and $\bar{Y_{\text{label}}}$ represent the means of $Y_{\text{pred}}$ and $Y_{\text{label}}$, respectively.

Structural Similarity Index (SSIM) For each local window or region, the SSIM is calculated as:

$$\text{SSIM}(x, y) = \frac{(2\mu_x\mu_y + C_1)(2\sigma_{xy} + C_2)}{(\mu_x^2 + \mu_y^2 + C_1)(\sigma_x^2 + \sigma_y^2 + C_2)} \tag{48}$$

where $x$ and $y$ are pixel values within two windows or regions, $\mu_x$ and $\mu_y$ are their means, $\sigma_x^2$ and $\sigma_y^2$ are their variances, and $\sigma_{xy}$ is their covariance. $C_1$ and $C_2$ are small constants to prevent division by zero.

Peak Signal-to-Noise Ratio (PSNR) The PSNR is given by:

$$\text{PSNR} = 10 \times \log_{10}\left(\frac{\text{MAX}^2}{\text{MSE}}\right) \tag{49}$$

where MAX represents the maximum possible pixel value. For an 8-bit image, MAX $= 255$.

## C.2 HYPERPARAMETERS

In the experimental settings, various hyperparameters are set for different datasets. For the attention head, the Turbulence, KTH, and Video DS datasets are set to 2, while the ERA5, SEVIR, and FLAME FLOW datasets are set to 4. The Fourier Transform Layers are configured as follows: 6 for Turbulence, ERA5, and SEVIR; 4 for FLAME FLOW; 10 for KTH; and 12 for Video DS. The hidden layer dimension in both the Feature Selection Module (FSM) and Other Training Module (OTM) is set to 64 for Turbulence, ERA5, and KTH, and 128 for SEVIR, FLAME FLOW, and Video DS. Across all datasets, the learning rate is consistently set at 0.01. In terms of the number of epochs, Turbulence, ERA5, SEVIR, and KTH have 500 epochs, FLAME FLOW has 300, and Video DS has 100. Lastly, the batch sizes vary: 2 for Turbulence, 6 for ERA5, 10 for both SEVIR and KTH, 4 for FLAME FLOW, and 20 for Video DS.

Table 3: Hyperparameters for Different Datasets

| Hyperparameter | Turbulence | ERA5 | SEVIR | FLAME FLOW | KTH | Video DS |
|---|---|---|---|---|---|---|
| Attention head | 2 | 4 | 4 | 4 | 2 | 2 |
| Fourier Transform Layers | 6 | 6 | 6 | 4 | 10 | 12 |
| Hidden layer dimension in FSM | 64 | 64 | 128 | 128 | 64 | 128 |
| Hidden layer dimension in OTM | 64 | 64 | 128 | 128 | 64 | 128 |
| Learning rate | 0.01 | 0.01 | 0.01 | 0.01 | 0.01 | 0.01 |
| Number of epochs | 500 | 500 | 500 | 300 | 500 | 100 |
| Batch size | 2 | 6 | 10 | 4 | 10 | 20 |

## D ADDITIONAL EXPERIMENTS

Table 4: Performance of Models on Various Datasets

| MODEL | Datasets | | | | | | | | |
|---|---|---|---|---|---|---|---|---|---|
| | CMS | RDS | SPS | RDPS | EDPS | SSS | ADS | LLS | FS |
| ADFV | 87.24 | 90.93 | 94.78 | 93.43 | 91.32 | 87.74 | 88.65 | 82.41 | 92.87 |
| STAC | 88.64 | 92.14 | 96.34 | 95.26 | 93.14 | 88.96 | 90.12 | 83.90 | 94.54 |

Our dataset is derived from (Chen et al., 2022), nine dynamics system datasets, recorded in video format. Comparison experiment with the model of the original text, we named the original text model as ADFV, because it is a video recording, we use SSIM as the evaluation metrics, and the experimental results are shown in the Table 4.

According to the experimental results, the STAC model performs better on all datasets compared to the original ADVF model. Specifically, the SSIM scores of the STAC model are higher than those of the ADVF model both on the CMS, RDS, SPS, RDPS, EDPS, SSS, ADS, LLS, and FS datasets, which clearly highlights the advantages and efficiency of the STAC model. These results demonstrate the strong performance and reliability of the STAC model in dealing with dynamic systems of video recordings.

## E LONG-TERM PREDICTION RESULTS OF STAC

In this section, we present the complete visualization results of STAC on the long-term prediction benchmark. We observe that on the Flame benchmark, our model is capable of excellently reconstructing the forecast results over an extended time frame, nearly encapsulating detailed contour information of the fire dynamics as well as the flow velocity. The astonishing consistency between the ground-truth and prediction at 210 seconds further substantiates our model's prowess in long-term forecasting.

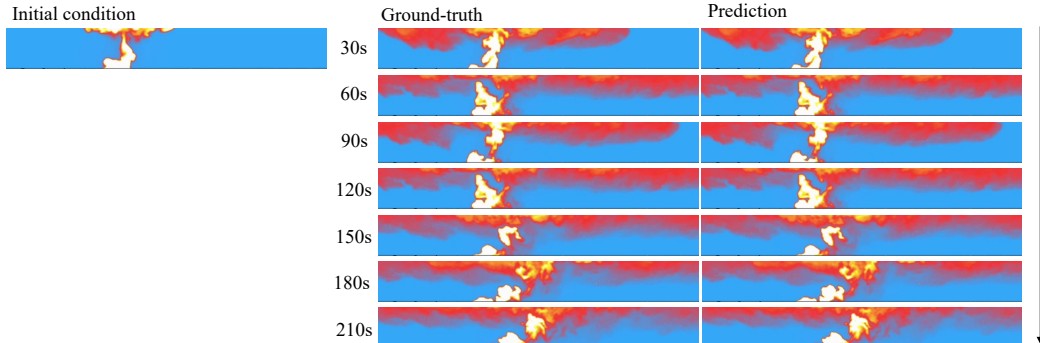

Figure 10: Case study of predicting the fire temperature field for the next 210 seconds based on the dynamic of the past 30 seconds.

## F ABLATION STUDY

### F.1 DATASET DETIALS

We conduct a comprehensive ablation study on STAC, with the dataset sourced from (Yin et al., 2022). The dataset is based on the shallow water equation, and our primary focus is on modeling its velocity field. The dataset has a basic dimensionality of [160, 1, 128, 256], which signifies 160 consecutive time snapshots. The 1 in the dimension stands for the channel variable, representing velocity. The resolution of the velocity field is captured by the 128x256 dimension. We slice the dataset into two sections: [20,1,128,256] and [140,1,128,256]. The former serves as the input, while the latter acts as the ground truth. Subsequently, these slices are fed into various model variants for ablation experiments. In addition, for the completeness of the ablation experiment, Fire and KTH datasets are added, and MSE and SSIM are selected as evaluation indicators.

### F.2 EXPERMENTAL RESULTS

We have designed the ablation experiments, and the specific model variants are shown below:

1. **STAC w/o FNO:** Indicates that STAC removed the FNO component.
2. **STAC w/o Transformer:** Indicates that STAC removed the Transformer component.
3. **STAC w/o Transformer:** Indicates that STAC removed the Transformer component.
4. **STAC w/o FSM:** Indicates that STAC removed the FSM component.
5. **STAC (OTM → ConvLSTM):** STAC replaces the OTM module with ConvLSTM.
6. **STAC (OTM → PredRNN):** STAC replaces the OTM module with PredRNN.
7. **STAC w/o OTM:** Indicates that STAC removed the OTM component.
8. **STAC w/o IFTM:** Indicates that STAC removed the IFTM component.
9. **STAC (CRP → Recall gate):** STAC replaces the CRP module with Recall gate.
10. **STAC w/o IFTM:** Indicates that STAC removed the IFTM component.

The results of this experiment show that the original STAC model performs well across different datasets (e.g. Fire, KTH, SWE), showing high SSIM scores and low MSE, indicating its effectiveness in the field of physical dynamical systems modeling. The experiment also involved removing or replacing different components of the STAC model, such as FNO, Transformer, FSM, OTM, IFTM, and CRP. Removing these components usually results in performance degradation, indicating the importance of each component to the overall performance of the model. In particular, when replacing OTM modules, the use of ConvLSTM resulted in significant performance degradation, while the use of PredRNN resulted in out-of-memory errors, suggesting inapplicability or requiring more optimization or resources. In addition, different model variants also differ in computation time, showing the tradeoff between model complexity, computational efficiency, and performance. Overall, the original

STAC model outperformed its variants in almost every respect, highlighting the importance of each component and balancing model complexity with efficiency in practical applications.

Table 5: Model Performance Metrics

| Model | Fire | | KTH | | SWE | | Avg Time |
|---|---|---|---|---|---|---|---|
| | MSE | SSIM | MSE | SSIM | MSE | SSIM | |
| STAC | 0.0487 | 92.87 | 0.2983 | 92.83 | 0.824 | 97.68 | 242s |
| STAC w/o FNO | 0.0756 | 84.54 | 0.3021 | 90.79 | 2.235 | 79.43 | 239s |
| STAC w/o Transformer | 0.0506 | 91.21 | 0.4765 | 83.72 | 1.093 | 94.32 | 192s |
| STAC w/o FSM | 0.0765 | 85.43 | 0.4893 | 82.9 | 2.352 | 76.54 | 187s |
| STAC (OTM → ConvLSTM) | 0.1723 | 65.69 | 1.2043 | 65.42 | 1.845 | 83.43 | 321s |
| STAC (OTM → PredRNN) | 0.3922 | 57.98 | OOM | OOM | OOM | OOM | 654s |
| STAC w/o OTM | 0.0802 | 82.33 | 0.5644 | 78.43 | 2.153 | 79.97 | 209s |
| STAC w/o IFTM | 0.0596 | 90.33 | 0.7321 | 70.43 | 1.183 | 91.23 | 237s |
| STAC (CRP → Recall gate) | 0.0998 | 79.49 | 0.5145 | 80.39 | 5.986 | 54.32 | 598s |
| STAC w/o CRP | 0.0543 | 90.83 | 0.3343 | 87.38 | 6.322 | 45.65 | 200s |

## G STATISTICAL DESCRIPTION OF DATASETS

A complete statistical description of the data set used in this paper is shown in the Table. 6

Table 6: Dataset statistics. $N\_tr$ and $N\_te$ denote the number of instances in the training and test sets. The lengths of the input and prediction sequences are $I\_l$ and $O\_l$, respectively.

| Dataset | N_tr | N_te | (C, H, W) | I_l | O_l | Interval |
|---|---|---|---|---|---|---|
| Turbulence | 5000 | 1000 | (3, 300, 300) | 50 | 50 | 1 second |
| ERA5 (Global) | 10000 | 2000 | (1, 1440, 720) | 10 | 10 | 1 day |
| ERA5 (Local) | 5000 | 1000 | (2, 200, 200) | 8 | 8 | 1 hour |
| KTH | 108717 | 4086 | (1, 128, 128) | 10 | 20 | 1 step |
| SEVIR | 4158 | 500 | (1, 384, 384) | 13 | 12 | 5 mins |
| Fire | 6000 | 1500 | (2, 32, 480) | 50 | 350 | 1 second |
| SWE | 2000 | 200 | (1, 128, 256) | 20 | 140 | 1 second |
| dynamics system | 6000 | 1200 | (3, 128, 128) | 2 | 2 | 1 second |

## H COMPLEXITY ANALYSIS

The complexity of the methods proposed in this paper is compared as follows. As can be seen from the results, the efficiency of our method is competitive.

Table 7: Complexity analysis

| Model | Memory (MB) | FLOPs (G) | Params (M) | Training time |
|---|---|---|---|---|
| FNO | 8.41 | 12.31 | 7.271 | 32s / epoch |
| F-FNO | 12.3 | 13.12 | 11.21 | 76s / epoch |
| E3D-LSTM | 2691 | 288.9 | 51 | 172s / epoch |
| MIM | 2331 | 179.2 | 38 | 154s / epoch |
| PredRNN-V2 | 1721 | 117.3 | 23.9 | 126s / epoch |
| SimVP-V2 | 421 | 17.2 | 46.8 | 25s / epoch |
| LSM | 10.21 | 14.31 | 9.002 | 37 s / epoch |
| U-NO | 92 | 32.1 | 136 | 278 s / epoch |
| STAC | 578 | 22.81 | 25.4 | 98s / epoch |

# I    IMPACT OF LEARNABLE $t$

We investigate the question of whether t is learnable or not in the ODE module, and we try the learnable $t$ and compare its performance as follows. The results, as shown in the Table 8, show that a learnable $t$ can slightly improve performance and is more flexible than a fixed $t$.

Table 8: Comparative experiments on learnable $t$

| Model | MSE (Fire) | SSIM (Fire) | MSE(KTH) | SSIM (KTH) | MSE(SWE) | SSIM(SWE) |
|-------|-----------|-------------|----------|------------|----------|-----------|
| STAC | 0.048 | 92.87 | 0.298 | 92.83 | 0.824 | 97.68 |
| STAC (learnable $t$) | 0.046 | 93.02 | 0.287 | 93.11 | 0.814 | 98.53 |

