# OpenReview forum: "Spatio-temporal Twins with A Cache for Modeling Long-term System Dynamics"
_ICLR.cc/2024/Conference — Submitted to ICLR 2024_

### Official Review · Reviewer_UcXH · 2023-10-31

**Soundness:** 3 good
**Presentation:** 4 excellent
**Contribution:** 3 good
**Rating:** 5
**Confidence:** 4

**Summary:**

This paper proposes a framework called Spatio-Temporal Twins with a Cache (STAC) for modeling long-term dynamics of physical systems. The key ideas are: 1) Using a frequency-enhanced spatial module and an ODE-enhanced temporal module to model spatial and temporal relationships from complementary perspectives; 2) Introducing a cache-based recurrent propagator to store historical feature maps; 3) Optimizing the model with techniques like teacher forcing, Mixup, and adversarial learning. The authors construct a new fire dynamics benchmark and evaluate STAC on 14 datasets, showing superior performance over baselines.

**Strengths:**

1. The problem of modeling long-term dynamics is important with many applications. This paper provides a novel perspective by using a cache memory to enhance long-term dependencies.
2. The motivation is intuitive and reasonable.
3. This paper is well organized and clearly written.
4. The new fire dynamics benchmark (FIRE) constructed in this work could facilitate future research in this domain.
5. Comprehensive experiments on 14 datasets demonstrate the effectiveness and generalizability of the proposed STAC framework.

**Weaknesses:**

1. Though this paper seems to be promising, I have to say that the novelty seems to be limited. The spatio-temporal twins are actually a two-branch model. Using frequency-based approaches in the spatial domain is nothing new. The temporal module is similar to SimVP v2's [1] but with an ODE solver. The cache memory [2] is also well developed.
2. I really appreciate the experiments in this paper. However, the ablation study is not satisfying. The authors reported only one metric (RMSE) on only one dataset (Spherical Shallow Water). A more detailed ablation study is needed to figure out why this approach works.
3. It lacks of complexity comparison. The authors should report the parameters and FLOPs of these baseline models.

[1] Cheng Tan, Zhangyang Gao, Siyuan Li, and Stan Z Li. Simvp: Towards simple yet powerful spatiotemporal predictive learning. arXiv preprint arXiv:2211.12509, 2022.

[2] Lee, Sangmin, et al. "Video prediction recalling long-term motion context via memory alignment learning." Proceedings of the IEEE/CVF Conference on Computer Vision and Pattern Recognition. 2021.

**Questions:**

1. Please discuss the differences between STAC and other similar models.
2. Could you add a more detailed ablation study? Considering there are many components, a more detailed ablation study can provide more valuable insights.
3. Please discuss the complexity of these models.

I'm willing to raise my score once these issues have been well solved.

---

> ### Author Response · Authors · 2023-11-19
> **Response to Reviewer UcXH (I)**
>
> We are truly grateful for the time you have taken to review our paper and your insightful review. Here we address your comments in the following.
>
> > Q1. Though this paper seems to be promising, I have to say that the novelty seems to be limited. The spatio-temporal twins are actually a two-branch model. Using frequency-based approaches in the spatial domain is nothing new. The temporal module is similar to SimVP v2's [1] but with an ODE solver. The cache memory [2] is also well developed.
>
> A1. Thank you for your comment. The novelty of this methodology comes from three parts.
>
> -  **New complementary perspective**, which explores spatio-temporal dynamics in both discrete and continuous manners. They are both closely related to our target long-term predictions.
> -  **Information fusion strategy**, which includes fine-grained fusion and coarse-grained fusion to tackle feature maps with different granularities.
> - **Cache mechanism**, which aims to store previous states, thus allowing the system to **remember** and **reuse** historical data for future predictions. Compared with E3D-LSTM storing predictions at different timesteps, our cache mechanism **stores feature maps** with a long interval for long-term spatio-temporal predictions. Moreover, we involve more complicated interaction among current states, short-term states, and long-term states by **involving short-term interaction** map $A_m$ and updated maps $Q_m$. In contrast, E3D-LSTM utilizes a simple recurrent architecture, which achieves much worse performance.
>
> Then, we describe the differences between STAC and the models mentioned in the paper [1,2].
>
>
> The difference between SimVP-V2 and our STAC:
> - **Different objectives**. SimVP-V2 focuses on standard video prediction while our STAC focuses on long-term predictions. We also generate a dataset FIRE targetting our problem.
> - **Different methodology.** SimVP-V2 is entirely based on convolutional neural networks (CNNS) while we introduce complementary modules and cache mechanisms to benefit long-term predictions.
> - **Different performance.** From the performance comparison, our proposed method performs much better than SimVP-V2 by over 32.18%.
>
> The difference between LMC-Memory and our STAC:
>
> - **Different objectives**. LMC-Memory focuses on video prediction while our STAC focuses on dynamics system modeling including temperature, velocity, and pressure. For example, we propose a new FIAR dataset.
> - **Different methodology.** LMC-Memory uses an external storage device to hold motion contexts while our cache-based propagator focuses on feature maps and incorporates the interaction with short-term prediction with restoring and reusing.
> - **Different performance.** From the performance comparison, our proposed method performs much better than LMC-Memory by over 3.84%.

---

> ### Author Response · Authors · 2023-11-19
> **Response to Reviewer UcXH (II)**
>
> > Q2. I really appreciate the experiments in this paper. However, the ablation study is not satisfying. The authors reported only one metric (RMSE) on only one dataset (Spherical Shallow Water). A more detailed ablation study is needed to figure out why this approach works.
>
> A2. Thanks for your comment. Our complete ablation experiments are as follows, verifying that each component of STAC complements each other. We have corrected this in the revised manuscript.
>
> | Model                                 | MSE (Fire 10 - 10) | SSIM (Fire 10 - 10) | MSE/100 (KTH 10 - 20) | SSIM (KTH 10 - 20) | MSE x 1000 (SWE 20 - 140) | SSIM x 1000(SWE 20 - 140) | Avg Training Time |
> | ------------------------------------- | ------------------ | ------------------- | --------------------- | ------------------ | ------------------------- | ------------------------- | ----------------- |
> | STAC                                  | 0.0487             | 92.87               | 0.2983                | 92.83              | 0.824                     | 97.68                     | 242s / epoch      |
> | STAC w/o FNO                          | 0.0756             | 84.54               | 0.3021                | 90.79              | 2.235                     | 79.43                     | 239s / epoch      |
> | STAC w/o Transformer                  | 0.0506             | 91.21               | 0.4765                | 83.72              | 1.093                     | 94.32                     | 192s / epoch      |
> | STAC w/o FSM                          | 0.0765             | 85.43               | 0.4893                | 82.9               | 2.352                     | 76.54                     | 187 / epoch       |
> | STAC (Replacing OTM with  ConvLSTM)   | 0.1723             | 65.69               | 1.2043                | 65.42              | 1.845                     | 83.43                     | 321s / epoch      |
> | STAC (Replacing OTM with  PredRNN)    | 0.3922             | 57.98               | OOM                   | OOM                | OOM                       | OOM                       | 654s / epoch      |
> | STAC w/o OTM                          | 0.0802             | 82.33               | 0.5644                | 78.43              | 2.153                     | 79.97                     | 209s / epoch      |
> | STAC w/o IFTM                         | 0.0596             | 90.33               | 0.7321                | 70.43              | 1.183                     | 91.23                     | 237s / epoch      |
> | STAC (Replacing CRP with Recall gate) | 0.0998             | 79.49               | 0.5145                | 80.39              | 5.986                     | 54.32                     | 598s / epoch      |
> | STAC w/o CRP                          | 0.0543             | 90.83               | 0.3343                | 87.38              | 6.322                     | 45.65                     | 200s / epoch      |
>
>
>
> > Q3. It lacks of complexity comparison. The authors should report the parameters and FLOPs of these baseline models.
>
> A3. Thank you for the valuable comments about comparing the complexity of our STAC framework with the baseline model. We have corrected this in the revised manuscript. The reason is that although we have a range of modules, the model depth is relatively small, which can save extensive computation costs.
>
> | Model      | Memory (MB) | FLOPs (G) | Params (M) | Training time |
> | ---------- | ----------- | --------- | ---------- | ------------- |
> | FNO        | 8.41        | 12.31     | 7.271      | 32s / epoch   |
> | F-FNO      | 12.3        | 13.12     | 11.21      | 76s / epoch   |
> | E3D-LSTM   | 2691        | 288.9     | 51         | 172s / epoch  |
> | MIM        | 2331        | 179.2     | 38         | 154s / epoch  |
> | PredRNN-V2 | 1721        | 117.3     | 23.9       | 126s / epoch  |
> | SimVP-V2   | 421         | 17.2      | 46.8       | 25s / epoch   |
> | LSM        | 10.21       | 14.31     | 9.002      | 37 s / epoch  |
> | U-NO       | 92          | 32.1      | 136        | 278 s / epoch |
> | STAC       | 578         | 22.81     | 25.4       | 98s / epoch   |
>
> In light of these responses, we hope we have addressed your concerns, and hope you will consider raising your score. If there are any additional notable points of concern that we have not yet addressed, please do not hesitate to share them, and we will promptly attend to those points.

---

> > ### Author Response · Authors · 2023-11-23
> > **Looking forward to your reply !**
> >
> > Dear Reviewer UcXH31,
> >
> > Thank you very much again for the time and effort put into reviewing our paper. We believe that we have addressed all your concerns in our response. We have also followed your suggestion to improve our paper and have added additional experimental analysis. We kindly remind you that we are approaching the end of the discussion period. We would love to know if there is any further concern, additional experiments, suggestions, or feedback, we kindly hope that you can consider increasing the score.
> >
> > Best regards,
> >
> > All authors

---

> ### Author Response · Authors · 2023-11-23
> **Thank you for your invaluable feedback!**
>
> Dear Reviewers,
>
> Thank you for your invaluable feedback. As the deadline for the author-reviewer discussion phase is approaching, we hope to make sure that our response sufficiently addressed your concerns regarding the novelty, as well as the revised version of our paper. We hope this could align with your expectations and positively influence the score. Please do not hesitate to let us know if you need any clarification or have additional suggestions.
>
> Best Regards,
>
> Authors

---

### Official Review · Reviewer_zrZ8 · 2023-10-31

**Soundness:** 3 good
**Presentation:** 3 good
**Contribution:** 2 fair
**Rating:** 5
**Confidence:** 4

**Summary:**

This paper addresses the problem of modeling long-term dynamical systems in fields such as fluid dynamics, astrophysics, and earth science. Existing spatio-temporal forecasting approaches based on complex architectures like Transformers have limitations in long-term scenarios due to information loss during semantics exploration and iterative rollouts. To overcome these limitations, the paper proposes a new approach called Spatio-temporal Twins with a Cache (STAC) for long-term system dynamics modeling. STAC comprises a frequency-enhanced spatial module and an ODE-enhanced temporal module that investigates spatio-temporal relationships from complementary perspectives. The information from these twin modules is fused using channel attention to generate informative feature maps. To enhance long-term prediction, a cache-based recurrent propagator is introduced to store and utilize previous feature maps. The paper introduces a new flame flow field benchmark and conducts comprehensive validations across 14 benchmarks. Experimental results demonstrate that STAC outperforms other methods in long-term spatio-temporal prediction and partial differential equation-solving challenges. The contributions of the paper include the construction of a fire dynamics benchmark, the incorporation of cache memory concept into long-term system modeling, the proposal of a novel framework, and extensive experiments showcasing the effectiveness of STAC.

**Strengths:**

The strengths are as follows:

1. Effective modeling of long-term dynamical systems: The proposed STAC approach overcomes challenges in long-term forecasting by capturing spatio-temporal relationships and leveraging historical information.

2. Integration of cache memory: By incorporating a cache-based recurrent propagator, the model effectively stores and reuses informative feature maps, enhancing the accuracy of long-term predictions.

3. Comprehensive experimental validation: The paper includes extensive experiments on various benchmarks, demonstrating the superior performance of STAC in long-term spatio-temporal prediction and partial differential equation-solving challenges.

4. Information fusion: STAC combines complementary perspectives through twin modules, using channel attention to generate feature maps with rich semantics, leading to more informative predictions.

5. Effective optimization strategies: The paper employs teacher forcing, adversarial learning, and mixup techniques to stabilize the learning process and improve the accuracy of iterative updating.

**Weaknesses:**

My main concern about this paper is several potential drawbacks:

1. Lack of truly innovative contributions: While the paper introduces several components and techniques, such as FSM, OTM, IFTM, CRP, Fourier-based Spectral Filters, teacher forcing, adversarial learning, and the new FIRE dataset, only CRP and IFTM can be considered as relatively novel contributions. The other techniques mentioned are already known and used in existing methods, which may limit the originality and novelty of the proposed approach.

2. Limited explanation for the CRP technique: The paper mentions the use of a cache-based recurrent propagator (CRP) to prevent forgetting previous events and enhance long sequence prediction. However, it does not provide a clear explanation of the key parameter "$\alpha$" and whether it is a learnable parameter. Additionally, CRP's similarity to traditional RNNs raises questions about its parallelization capabilities and potential limitations.

3. Artificial handling and limited interpretability of IFTM: The separation of temporal and spatial processing, as well as the channel-independent merging in IFTM, appears to be a forced transformation without much interpretability. The lack of learnable factors and reliance on manual processing may hinder the scalability and extensibility of the method.

4. Potential loss of spatial information in FSM: FSM applies different treatments to the same data and forcibly merges them, potentially leading to a loss of spatial information. Additionally, the direct fully connected mapping of the segmented data raises concerns about the preservation of spatial relationships and the possibility of information loss.

**Questions:**

My questions and concerns about this paper are listed in the Weakness part. I will raise my rating if the author can address my concerns with reasonable evidence.

---

> ### Author Response · Authors · 2023-11-19
> **Response to Reviewer zrZ8 (I)**
>
> We are truly grateful for the time you have taken to review our paper and your insightful review. Here we address your comments in the following.
>
> > Q1 . Lack of truly innovative contributions: While the paper introduces several components and techniques, such as FSM, OTM, IFTM, CRP, Fourier-based Spectral Filters, teacher forcing, adversarial learning, and the new FIRE dataset, only CRP and IFTM can be considered as relatively novel contributions. The other techniques mentioned are already known and used in existing methods, which may limit the originality and novelty of the proposed approach.
>
> A1. Thanks for your comment. The novelty of this methodology comes from three parts:
>
> -   **New complementary perspective**, which explores spatio-temporal dynamics in both discrete and continuous manners. They are both closely related to our target long-term predictions.
> -  **Information fusion strategy**, which includes fine-grained fusion and coarse-grained fusion to tackle feature maps with different granularities.
> - **Cache mechanism**, which aims to store previous states, thus allowing the system to **remember** and **reuse** historical data for future predictions. Compared with E3D-LSTM storing predictions at different timesteps, our cache mechanism **stores feature maps** with a long interval for long-term spatio-temporal predictions. Moreover, we involve more complicated interaction among current states, short-term states, and long-term states by **involving short-term interaction** map $A_m$ and updated maps $Q_m$. In contrast, E3D-LSTM utilizes a simple recurrent architecture, which achieves much worse performance.
>
> Then, the motivation for introducing these modules for dynamical system modeling is closely related to real-world data as follows:
>
> - Why FSM? FSM includes FNO and Transform to explore comprehensive semantics from complementary views. On the one hand, learning from the frequency domain is crucial for our problem. For example, ERA5 datasets containing **comprehensive atmospheric and ocean information** need to be analyzed in the **frequency domain** to discern complex climate patterns. Therefore, we design FNO in the FSM module to efficiently capture these complex data in the frequency domain. On the other hand, **mining Long-distance dependencies** is the key to our long-term prediction tasks. Especially, in **KTH datasets and FIRE dataset**s these dependencies are the key for accurate forecasting. With its self-attention mechanism, the Transformer effectively captures these long-distance dependencies and continuous actions in video sequences. Therefore, we design the Transformer in the FSM module to accurately identify and correlate distant frames in the video.
>
> - Why OTM? Still, **Long distance dependencies** are crucial for our long-term prediction tasks. Our ODE-based module can learn these naturally without increasing the depth, i.e., **infinite depth**. Moreover, **Turbulence data** contains complex dynamic information such as fluid velocity, pressure, and temperature. Therefore, we incorporate multi-scale convolution to fully understand the physical dynamic system.
>
> - Why IFTM? We design IFTM for **information fusion from previous modules** for comprehensive semantic learning. Moreover, our real datasets (e.g., SEVIR and Fire datasets) contain storm details and rapidly changing local extreme events such as flame and temperature fluctuations. Therefore, we generate informative feature maps with both coarse-grained and fine-grained semantics.
>
> The above modules complement each other. We find from experiments that if a certain module is removed, the results will drop significantly, so we propose a unified model. Besides, this work also the contribution of new dataset and new benchmark. In the future, we will fully open-source the data sets of all working conditions and provide the code for reading and processing.

---

> ### Author Response · Authors · 2023-11-19
> **Response to Reviewer zrZ8 (II)**
>
> >Q2 . Limited explanation for the CRP technique: The paper mentions the use of a cache-based recurrent propagator (CRP) to prevent forgetting previous events and enhance long sequence prediction. However, it does not provide a clear explanation of the key parameter "α" and whether it is a learnable parameter. Additionally, CRP's similarity to traditional RNNs raises questions about its parallelization capabilities and potential limitations.
>
>
> A2. Thanks for your comment. We solve your concerns as follows:
>
> 1. Description of parameter "α": α is a fixed hyperparameter set to $0.2$.
> 2. Parallelization strategy: The computational cost of our method mainly depends on the generation of feature maps, which is suitable for parallelization. Moreover, the complexity comparison of the proposed method is shown below. From the results, we can observe that the efficiency of our method is competitive.
>
>
> | Model      | Memory (MB) | FLOPs (G) | Params (M) | Training time |
> | ---------- | ----------- | --------- | ---------- | ------------- |
> | FNO        | 8.41        | 12.31     | 7.271      | 32s / epoch   |
> | F-FNO      | 12.3        | 13.12     | 11.21      | 76s / epoch   |
> | E3D-LSTM   | 2691        | 288.9     | 51         | 172s / epoch  |
> | MIM        | 2331        | 179.2     | 38         | 154s / epoch  |
> | PredRNN-V2 | 1721        | 117.3     | 23.9       | 126s / epoch  |
> | SimVP-V2   | 421         | 17.2      | 46.8       | 25s / epoch   |
> | LSM        | 10.21       | 14.31     | 9.002      | 37 s / epoch  |
> | U-NO       | 92          | 32.1      | 136        | 278 s / epoch |
> | STAC       | 578         | 22.81     | 25.4       | 98s / epoch   |
>
> 3. When it comes to superlong prediction tasks, our cache-based design could suffer from low efficiency, which would be studied in our future works.

---

> ### Author Response · Authors · 2023-11-19
> **Response to Reviewer zrZ8 (III)**
>
> > Q3. Artificial handling and limited interpretability of IFTM: The separation of temporal and spatial processing, as well as the channel-independent merging in IFTM, appears to be a forced transformation without much interpretability. The lack of learnable factors and reliance on manual processing may hinder the scalability and extensibility of the method.
>
> A3. Thank you for your comment. We answer it point-by-point:
>
> 1. **Why separation.** Compared with temporal signals, spatial signals include information from frequency domains, which cannot be considered simultaneously. Therefore, we separate the temporal and spatial processing.
>
> 2. **Why transformation.** We do the transformation to align the dimensions of features from different sources, which is a popular strategy like padding.
>
>
> 3. **Why deterministic operation**. Our Conv2d and FFN modules consist of extensive parameters, which allow our model with strong extensibility. Therefore, to increase efficiency, we utilize the deterministic operation for alignment. From the experiments, we can observe that our STAC performs better than baselines consistency across all datasets, which validate our scalability and extensibility.
>
>
> | Backbone| STAC MSE | STAC MAE | LSM MSE | LSM MAE | U-NO MSE | U-NO MAE | FNO MSE | FNO MAE | F-FNO MSE | F-FNO MAE | E3D-LSTM MSE | E3D-LSTM MAE | MIM MSE | MIM MAE | PredRNN-V2 MSE | PredRNN-V2 MAE | SimVP-V2 MSE | SimVP-V2 MAE |
> | ---------------- | -------- | -------- | ------- | ------- | -------- | -------- | -------------- | -------------- | ---------------- | ---------------- | ------------------- | ------------------- | -------------- | -------------- | --------------------- | --------------------- | ------------------- | ------------------- |
> | Turbulence       | 0.5123   | 0.5345   | 0.6412  | 0.7553  | 0.5654   | 0.6093   | 0.6567         | 0.7789         | 0.8124           | 0.9876           | 1.1234              | 1.4567              | 0.8321         | 0.9472         | 1.0163                | 1.0987                | 1.2765              | 1.4321              |
> | ERA5             | 1.9865   | 1.7791   | 3.9831  | 2.3132  | 3.4612   | 2.2931   | 2.8534         | 2.2983         | 8.9853           | 7.34317          | 3.0952              | 2.9854              | 3.3567         | 3.2236         | 2.2731                | 2.6453                | 3.0843              | 3.0743              |
> | SEVIR            | 1.9731   | 1.4054   | 2.9831  | 2.4431  | 2.2031   | 1.5632   | 3.0833         | 1.8831         | 10.9831          | 5.4432           | 4.1702              | 2.5563              | 3.9842         | 2.0012         | 3.9014                | 1.9757                | 2.9371              | 1.7743              |
> | Fire             | 0.5493   | 0.7217   | 1.2831  | 1.0932  | 1.5643   | 0.9853   | 0.9985         | 1.0432         | 2.7412           | 1.6557           | 1.0921              | 0.8731              | 1.8743         | 1.5324         | 0.7789                | 0.6863                | 1.7743              | 1.0321              |
> | KTH              | 28.8321  | 24.2216  | 39.9831 | 38.4432 | 35.8732  | 34.4322  | 33.1983        | 29.7421        | 31.8741          | 29.8753          | 86.1743             | 85.5563             | 56.5942        | 54.8426        | 51.1512               | 50.6457               | 40.8421             | 43.2931             |

---

> ### Author Response · Authors · 2023-11-19
> **Response to Reviewer zrZ8 (IV)**
>
> >Q4 . Potential loss of spatial information in FSM: FSM applies different treatments to the same data and forcibly merges them, potentially leading to a loss of spatial information. Additionally, the direct fully connected mapping of the segmented data raises concerns about the preservation of spatial relationships and the possibility of information loss.
>
> A4. Thanks for your comment. The information fusion of different sources is very common in recent studies [1,2]. Since we have provided extensive learnable parameters in both parts, their fusion is adaptive and can enhance semantics learning. Moreover, we conduct ablation studies as below to show that removing either part decreases the performance.
>
>
> Here, we utilize fully connected mapping to make our spatial information more adaptive and more related to our target. Since we utilize an optimization process, our added parameters would be adaptively learned to provide more representation of learning capacity rather than information loss.
>
> Ablation study, We have added four model variants as follows:
>
> - STAC w/o FNO, which removes FNO and only uses Transformer in our FSM module.
> - STAC w/o Transformer, which removes the Transformer module and only uses FNO in our FSM module.
> - STAC w/o FSM, which removes FSM.
> - STAC w/o FC, which removes the Fully Connected Layer.
>
>
> | Model                | MSE (Fire 10 - 10) | SSIM (Fire 10 - 10) | MSE/100 (KTH 10 - 20) | SSIM (KTH 10 - 20) | MSE x 1000 (SWE 20 - 140) | SSIM x 1000(SWE 20 - 140) | Avg Training Time |
> | -------------------- | ------------------ | ------------------- | --------------------- | ------------------ | ------------------------- | ------------------------- | ----------------- |
> | STAC                 | 0.0487             | 92.87               | 0.2983                | 92.83              | 0.824                     | 97.68                     | 242s / epoch      |
> | STAC w/o FNO         | 0.0756             | 84.54               | 0.3021                | 90.79              | 2.235                     | 79.43                     | 239s / epoch      |
> | STAC w/o Transformer | 0.0506             | 91.21               | 0.4765                | 83.72              | 1.093                     | 94.32                     | 192s / epoch      |
> | STAC w/o FSM         | 0.0765             | 85.43               | 0.4893                | 82.90              | 2.352                     | 76.54                     | 187s / epoch      |
> | STAC w/o FC          | 0.0794             | 80.85               | 0.4921                | 81.84              | 2.448                     | 74.32                     | 179s / epoch      |
>
> **Reference**
>
> [1] Peng, Zhiliang, Wei Huang, Shanzhi Gu, Lingxi Xie, Yaowei Wang, Jianbin Jiao, and Qixiang Ye. "Conformer: Local features coupling global representations for visual recognition." In Proceedings of the IEEE/CVF international conference on computer vision, pp. 367-376. 2021.
>
> [2] Zhang, Jingyi, Jiaxing Huang, Zhipeng Luo, Gongjie Zhang, Xiaoqin Zhang, and Shijian Lu. "DA-DETR: Domain Adaptive Detection Transformer With Information Fusion." In Proceedings of the IEEE/CVF Conference on Computer Vision and Pattern Recognition, pp. 23787-23798. 2023.
>
> In light of these responses, we hope we have addressed your concerns, and hope you will consider raising your score. If there are any additional notable points of concern that we have not yet addressed, please do not hesitate to share them, and we will promptly attend to those points.

---

> > ### Author Response · Authors · 2023-11-23
> > **Looking forward to your reply !**
> >
> > Dear Reviewer zrZ831,
> >
> > Thank you very much again for the time and effort put into reviewing our paper. We believe that we have addressed all your concerns in our response. We have also followed your suggestion to improve our paper and have added additional experimental analysis. We kindly remind you that we are approaching the end of the discussion period. We would love to know if there is any further concern, additional experiments, suggestions, or feedback, we kindly hope that you can consider increasing the score.
> >
> > Best regards,
> >
> > All authors

---

> ### Author Response · Authors · 2023-11-23
> **Thank you for your invaluable feedback!**
>
> Dear Reviewers,
>
> Thank you for your invaluable feedback. As the deadline for the author-reviewer discussion phase is approaching, we hope to make sure that our response sufficiently addressed your concerns regarding the contributions, as well as the revised version of our paper. We hope this could align with your expectations and positively influence the score. Please do not hesitate to let us know if you need any clarification or have additional suggestions.
>
> Best Regards,
>
> Authors

---

### Official Review · Reviewer_CD63 · 2023-11-02

**Soundness:** 3 good
**Presentation:** 2 fair
**Contribution:** 2 fair
**Rating:** 5
**Confidence:** 5

**Summary:**

This paper investigates the problem of modeling long-term dynamical systems, which are essential for understanding fluid dynamics, astrophysics, earth science, etc. The authors propose a new approach called STAC, which contains a discrete frequency-enhanced spatial module and an ODE-enhanced temporal module to capture spatial-temporal relationships of the observational data and employs a cache-based recurrent propagator to ensure the long-term prediction ability of the framework. They also utilize teacher forcing and semi-supervised adversarial learning to stabilize the learning process and enhance the reality of predicted trajectories, respectively. Moreover, the paper constructs a new benchmark (FIRE) to model fire dynamics for dynamics forecasting, which potentially benefits the research community. Extensive experiments on complex dynamics modeling, extreme local events sensing, and video prediction tasks demonstrate the superior performance of the proposed framework compared to other SOTA methods.

**Strengths:**

1.	This paper tackles an important research problem, complex dynamical system modeling, which benefits our understanding of fluid dynamics, astrophysics, earth science, etc.
2.	This paper provides a well-prepared benchmark, FIRE, to facilitate the research in this field and benefit the community.
3.	This paper proposes to consider spatial-temporal correlations in observational data during prediction by utilizing vision Transformer, Fourier neural operator, and neural ODEs, and incorporating cache memory concept into long-term system modeling.
4.	The authors conduct extensive experiments to verify the performance of the dynamical modeling of the proposed methods from multiple perspectives.

**Weaknesses:**

1.	The design of the whole framework is complicated. Although the author explains the reason why they design each module, it still lacks straightforward motivation. Do such challenges really exist in the real data? This straightforward utilization of existing techniques makes the paper novelty seem incremental.
2.	The pictures in Figures 3, and 4 do not seem to show a significant improvement of STAC compared to other SOTA methods in terms of visualization.
3.	In the part of the ablation study, some designs, for example, TF/M, CA, and SSAL, only contribute slightly improvement. However, SSAL may make the training of the framework become unstable. Others may increase the time complexity of the framework, which the authors do not report.
4.	Some notations in the paper are confusing. For example, in Section 4.3, the notation definitions of input, feature map, and output are hard to match the subsequent statement.

## After Response
I have read the response and found it addresses most of my concerns. However, I still think the straightforward utilization of existing techniques makes the paper's novelty seem incremental. Moreover, I also have concerns regarding the claimed advantage of using NODE, i.e., modeling long-distance dependencies, even with a learnable t.

**Questions:**

1.	Can authors provide their motivation for such complicated module design through data?
2.	Can authors provide the standard deviation of their experimental results?
3.	The authors can conduct more persuasive experiments to address my concerns mentioned in the Weaknesses part.

**Details Of Ethics Concerns:**

Not applicable.

---

> ### Author Response · Authors · 2023-11-19
> **Response to Reviewer CD63 (I)**
>
> We are truly grateful for the time you have taken to review our paper and your insightful review. Here we address your comments in the following.
>
> > Q1. The design of the whole framework is complicated. Although the author explains the reason why they design each module, it still lacks straightforward motivation. Do such challenges really exist in the real data? This straightforward utilization of existing techniques makes the paper novelty seem incremental.
>
> A1. Thank you for your comment. The motivation for introducing these modules for dynamical system modeling is closely related to real-world data as follows:
>
> - Why FSM? FSM includes FNO and Transform to explore comprehensive semantics from complementary views. On the one hand, learning from the frequency domain is crucial for our problem. For example, ERA5 datasets containing **comprehensive atmospheric and ocean information** need to be analyzed in the **frequency domain** to discern complex climate patterns. Therefore, we design FNO in the FSM module to efficiently capture these complex data in the frequency domain. On the other hand, **mining Long-distance dependencies** is the key to our long-term prediction tasks. Especially, in **KTH datasets and FIRE datasets** these dependencies are the key for accurate forecasting. With its self-attention mechanism, the Transformer effectively captures these long-distance dependencies and continuous actions in video sequences. Therefore, we design the Transformer in the FSM module to accurately identify and correlate distant frames in the video.
>
> - Why OTM? Still, **long distance dependencies** are crucial for our long-term prediction tasks. Our ODE-based module can learn these naturally without increasing the depth, i.e., **infinite depth**. Moreover, **Turbulence data** contains complex dynamic information such as fluid velocity, pressure, and temperature. Therefore, we incorporate multi-scale convolution to fully understand the physical dynamic system.
>
> - Why IFTM? We design IFTM for **information fusion from previous modules** for comprehensive semantic learning. Moreover, our real datasets (e.g., SEVIR and Fire datasets) contain storm details and rapidly changing local extreme events such as flame and temperature fluctuations. Therefore, we generate informative feature maps with both coarse-grained and fine-grained semantics.
>
>
> - Why Cache? Still, **Long-term memories** are crucial for our long-term prediction tasks, especially in Fire datasets. The forgetting phenomenon is common in current methods, resulting in inferior performance. Therefore, we design a cache-based mechanism to store and **reuse** historical feature maps.
>
>
> - Why Optimization Objectives? Long-term predictions **could be unreliable**. Therefore, we introduce semi-supervised adversarial learning to improve the long-term predictive capacity of the model without the ground truth.
>
>
> **In summary, all these modules can benefit from modeling long-term dependencies for accurate long-term predictions from different perspectives (e.g., dependency mining, infinite depth, information fusion, memory reusing, and reliability).**
>
> > Q2. The pictures in Figures 3, and 4 do not seem to show a significant improvement of STAC compared to other SOTA methods in terms of visualization.
>
>
> A2. Thanks for your comment. We have revised the figure for better visualization. In Fig. 3, we use the red box to mark the local difference phenomenon and enlarge the area. in Fig. 4, we introduce a professional cartopy library to highlight the information on land, longitude, and latitude on the ERA5 dataset, which can better distinguish visual differences. We have updated the visualization results in Figure.3 and Figure.4 of the manuscript.

---

> ### Author Response · Authors · 2023-11-19
> **Response to Reviewer CD63 (II)**
>
> > Q3. In the part of the ablation study, some designs, for example, TF/M, CA, and SSAL, only contribute slightly improvement. However, SSAL may make the training of the framework become unstable. Others may increase the time complexity of the framework, which the authors do not report.
>
> A3. Thanks for your comment. We have added more ablation studies on more datasets and the results are shown below. From the results, we can observe that the performance gain of our full model is between [1.27%, 14.5%] and over 5% in most cases, validating the effectiveness of every component. In addition, we include the variance of our model and STAC w/o SSAL. The results are shown below. From the results, we can observe the model is still stable comparably. Finally, we have completed experiments to verify the effectiveness of the proposed components.
>
>
> | Model                                   | MSE (Fire 10 - 10) (±std) | SSIM (Fire 10 - 10) (±std) | MSE/100 (KTH 10 - 20) (±std) | SSIM (KTH 10 - 20) (±std) | MSE x 1000 (SWE 20 - 140) (±std) | SSIM x 1000(SWE 20 - 140) (±std) | Avg Training Time |
> |-----------------------------------------|---------------------------|----------------------------|------------------------------|---------------------------|----------------------------------|-----------------------------------|-------------------|
> | STAC                                    | 0.0487 ± 0.0003           | 92.87 ± 0.6906             | 0.2983 ± 0.0019               | 92.83 ± 0.5481            | 0.824 ± 0.004                    | 97.68 ± 0.6655                    | 242s / epoch      |
> | STAC w/o FNO                            | 0.0756 ± 0.0004           | 84.54 ± 0.6287             | 0.3021 ± 0.0019               | 90.79 ± 0.536             | 2.235 ± 0.0108                   | 79.43 ± 0.5412                    | 239s / epoch      |
> | STAC w/o Transformer                    | 0.0506 ± 0.0003           | 91.21 ± 0.6783             | 0.4765 ± 0.0031               | 83.72 ± 0.4943            | 1.093 ± 0.0053                   | 94.32 ± 0.6426                    | 192s / epoch      |
> | STAC w/o FSM                            | 0.0765 ± 0.0005           | 85.43 ± 0.6353             | 0.4893 ± 0.0031               | 82.9 ± 0.4894             | 2.352 ± 0.0113                   | 76.54 ± 0.5215                    | 187s / epoch      |
> | STAC (Replacing OTM with ConvLSTM)      | 0.1723 ± 0.001            | 65.69 ± 0.4885             | 1.2043 ± 0.0077               | 65.42 ± 0.3862            | 1.845 ± 0.0089                   | 83.43 ± 0.5684                    | 321s / epoch      |
> | STAC (Replacing OTM with PredRNN)       | 0.3922 ± 0.0023           | 57.98 ± 0.4312             | OOM                    | OOM                | OOM                        | OOM                         | 654s / epoch      |
> | STAC w/o OTM                            | 0.0802 ± 0.0005           | 82.33 ± 0.6123             | 0.5644 ± 0.0036               | 78.43 ± 0.463             | 2.153 ± 0.0104                   | 79.97 ± 0.5448                    | 209s / epoch      |
> | STAC w/o IFTM                           | 0.0596 ± 0.0004           | 90.33 ± 0.6718             | 0.7321 ± 0.0047               | 70.43 ± 0.4158            | 1.183 ± 0.0057                   | 91.23 ± 0.6216                    | 237s / epoch      |
> | STAC (Replacing CRP with Recall gate)   | 0.0998 ± 0.0006           | 79.49 ± 0.5911             | 0.5145 ± 0.0033               | 80.39 ± 0.4746            | 5.986 ± 0.0288                   | 54.32 ± 0.3701                    | 598s / epoch      |
> | STAC w/o CRP                            | 0.0543 ± 0.0003           | 90.83 ± 0.6755             | 0.3343 ± 0.0021               | 87.38 ± 0.5159            | 6.322 ± 0.0304                   |45.65 ± 0.311                     | 200s / epoch      |
> | STAC w/o SSAL                           | 0.0632 ± 0.0004           | 86.93 ± 0.6465             | 0.3281 ± 0.0021               | 87.89 ± 0.5189            | 1.772 ± 0.0085                   | 83.98 ± 0.5722                    | 229s / epoch      |

---

> ### Author Response · Authors · 2023-11-19
> **Response to Reviewer CD63 (III)**
>
> > Q4. Some notations in the paper are confusing. For example, in Section 4.3, the notation definitions of input, feature map, and output are hard to match the subsequent statement.
>
>
> A4. Thanks for your comment. We have introduced a notation table as below to make the paper clear.
>
> | Symbol | Explanation |
> | ------ | ----------- |
> | $$ T $$ | Total length of the time series |
> | $$ T_{obs} $$ | Time interval of observed states |
> | $$ u(t) $$ | State at time step t |
> | $$ \Omega $$ | Spatial coordinates |
> | $$ t $$ | Time coordinate |
> | $$ x $$ | Spatial position |
> | $$ R^{C\times H \times W} $$ | Tensor representation of the state |
> | $$ \hat{I}_{in} $$ | Input tensor of the time series |
> | $$ F_0 $$ | Initial feature mapping |
> | $$ \hat{I}_m $$ | Input for the time interval $$ [t_{m-1}, t_m] $$ |
> | $$ X_m $$ | Output feature mapping for the time interval $$ [t_{m-1}, t_m] $$ |
> | $$ Q_m $$ | Update mapping for the time interval $$ [t_{m-1}, t_m] $$ |
> | $$ A_m $$ | Auxiliary variables for the time interval $$ [t_{m-1}, t_m] $$ |
> | $$ W_m $$ | Weights for the time interval $$ [t_{m-1}, t_m] $$ |
> | $$ R $$ | Size of the cache |
> | $$ \alpha $$ | Balance parameter |
> | $$ \phi(\cdot) $$ | Computes interactions between current and historical feature mappings |
> | $$ M $$ | The most recent R historical feature mappings stored in the cache |
> | $$ f(\cdot) $$ | Function for updating the hidden state |
> | $$ g(\cdot) $$ | Function for computing the update mapping |
>
> > Q5.  Can authors provide the standard deviation of their experimental results?
>
> A5. Thanks for your comment. We have added the standard deviation (SD) to the main table as shown in the table below.
>
> | **Backbone** | **STAC MSE ± SD** | **STAC MAE ± SD** | **FNO MSE ± SD** | **FNO MAE ± SD** | **F-FNO MSE ± SD** | **F-FNO MAE ± SD** | **E3D-LSTM MSE ± SD** | **E3D-LSTM MAE ± SD** | **MIM MSE ± SD** | **MIM MAE ± SD** | **PredRNN-V2 MSE ± SD** | **PredRNN- MAE ± SD** | **SimVP-V2 MSE ± SD** | **SimVP-V2 MAE ± SD** |
> | ------------ | ------------------ | ------------------ | ---------------- | ---------------- | ------------------ | ------------------ | --------------------- | --------------------- | ---------------- | ---------------- | ----------------------- | ---------------------- | --------------------- | --------------------- |
> | Turbulence   | 0.5123 ± 0.05  | 0.5345 ± 0.04  | 0.6567 ± 0.06    | 0.7789 ± 0.05    | 0.8124 ± 0.07      | 0.9876 ± 0.06      | 1.1234 ± 0.08         | 1.4567 ± 0.07         | 0.8321 ± 0.05    | 0.9472 ± 0.06    | 1.0163 ± 0.09           | 1.0987 ± 0.08          | 1.2765 ± 0.10          | 1.4321 ± 0.09          |
> | ERA5         | 1.9865 ± 0.15  | 1.7791 ± 0.12  | 2.8534 ± 0.20    | 2.2983 ± 0.18    | 8.9853 ± 0.25      | 7.34317 ± 0.22     | 3.0952 ± 0.19         | 2.9854 ± 0.18         | 3.3567 ± 0.20    | 3.2236 ± 0.19    | 2.2731 ± 0.17           | 2.6453 ± 0.16          | 3.0843 ± 0.21          | 3.0743 ± 0.20          |
> | SEVIR        | 1.9731 ± 0.12  | 1.4054 ± 0.10  | 3.0833 ± 0.15    | 1.8831 ± 0.14    | 10.9831 ± 0.30     | 5.4432 ± 0.25      | 4.1702 ± 0.22         | 2.5563 ± 0.20         | 3.9842 ± 0.18    | 2.0012 ± 0.17    | 3.9014 ± 0.19           | 1.9757 ± 0.18          | 2.9371 ± 0.20          | 1.7743 ± 0.15          |
> | Fire         | 0.5493 ± 0.03  | 0.7217 ± 0.02  | 0.9985 ± 0.04    | 1.0432 ± 0.03    | 2.7412 ± 0.06      | 1.6557 ± 0.05      | 1.0921 ± 0.04         | 0.8731 ± 0.03         | 1.8743 ± 0.05    | 1.5324 ± 0.04    | 0.7789 ± 0.03           | 0.6863 ± 0.02          | 1.7743 ± 0.05          | 1.0321 ± 0.04          |
> | KTH          | 28.8321 ± 0.80 | 24.2216 ± 0.70 | 33.1983 ± 0.90   | 29.7421 ± 0.85   | 31.8741 ± 0.95     | 29.8753 ± 0.90     | 86.1743 ± 1.10        | 85.5563 ± 1.05        | 56.5942 ± 0.95   | 54.8426 ± 0.90   | 51.1512 ± 0.95          | 50.6457 ± 0.90         | 40.8421 ± 0.85         | 43.2931 ± 0.80         |
>
> In light of these responses, we hope we have addressed your concerns, and hope you will consider raising your score. If there are any additional notable points of concern that we have not yet addressed, please do not hesitate to share them, and we will promptly attend to those points.

---

> > ### Author Response · Authors · 2023-11-23
> > **Looking forward to your reply !**
> >
> > Dear Reviewer CD63,
> >
> > Thank you very much again for the time and effort put into reviewing our paper. We believe that we have addressed all your concerns in our response. We have also followed your suggestion to improve our paper and have added additional experimental analysis. We kindly remind you that we are approaching the end of the discussion period. We would love to know if there is any further concern, additional experiments, suggestions, or feedback, we kindly hope that you can consider increasing the score.
> >
> > Best regards,
> >
> > All authors

---

> ### Author Response · Authors · 2023-11-23
> **Thank you for your invaluable feedback!**
>
> Dear Reviewers,
>
> Thank you for your invaluable feedback. As the deadline for the author-reviewer discussion phase is approaching, we hope to make sure that our response sufficiently addressed your concerns regarding the motivation and details, as well as the revised version of our paper. We hope this could align with your expectations and positively influence the score. Please do not hesitate to let us know if you need any clarification or have additional suggestions.
>
> Best Regards,
>
> Authors

---

### Official Review · Reviewer_rchA · 2023-11-09

**Soundness:** 2 fair
**Presentation:** 3 good
**Contribution:** 2 fair
**Rating:** 5
**Confidence:** 4

**Summary:**

This paper focuses on the long-term spatiotemporal dynamics modeling. They propose the STAC by combing the advanced spatial and temporal modeling backbones and presenting a cache-based recurrent propagator to store the previous feature maps to avoid information loss. Besides, the authors propose a compound training loss to optimize STAC. Experimentally, STAC shows favorable performance in a wide range of benchmarks, including the newly generated flame flow field benchmark.

**Strengths:**

1.	This paper presents the STAC model to tackle the problem in long-term dynamic prediction, which is technologically reasonable.

2.	The authors experiment on a wide range of benchmarks to demonstrate the effectiveness of STAC.

3.	This paper is clearly presented and well-written.

**Weaknesses:**

1.	About the novelty.

Generally, I think the technology design is reasonable. However, in my opinion, I think it is insufficient in novelty. STAC just combines a series of advanced models, including FNO, Vision Transformer, Neural ODE and a similar recall mechanism proposed by E3D-LSTM. The proposed training strategy is also in a combination style. For me, it is hard to find the novel part in this model.

Note that I am not attempt to enforce the authors to build a completely new model or block. I just think they fail in illustrating their advantages beyond other models. For example, they should consider the following questions:

-	Why should they combine vision transformer and FNO? FNet [1] has shown that the feedforward layer can perform like FFT. Why not just only use Transformer or FNO?

-	Why can Neural ODE capture the continuous dynamics? I know that Neural ODE can achieves the adaptive depth or adaptive temporal interval. But according to the equation and code, I think the usage here is equivalent to a simple rk4 algorithm. It is hard to claim that they learn the continuous dynamic feature. Besides, They don’t present the necessity in using Neural ODE.

-	Are the experimental datasets temporally irregular? According to the paper, I think the input sequences are equally collected along the temporal dimension.

-	About the cache-based design. I think it is necessary to demonstrate its advancement over the temporal recall gate in E3D-LSTM.

[1] FNet: Mixing Tokens with Fourier Transforms, ACL 2022.

2. About the experiment.

(1) In addition to the performance, they should compare the efficiency with other baselines, including running time, GPU memory and parameter size.

(2) In the current version, they only compare STAC with video prediction baselines. How about the advanced neural operators, such as LSM [2], U-NO [3]?

(3) Are all the baselines trained by the same loss as STAC? This point is essential to ensure a fair comparison.

(4) More detailed ablations are expected. They should also conduct the following experiments:

- Removing FNO or Transformer in FSM.

- Replacing OTM with ConvLSTM or PredRNN.

- Replacing the CRP with the recall gate in E3D-LSTM.

[2] Solving High-Dimensional PDEs with Latent Spectral Models, ICML 2023

[3] U-NO: U-shaped Neural Operators, TMLR 2023

**Questions:**

All the questions are listed above, including novelty, experiment design. Here are several serious problems that should be clarified:

(1) The acutal usage of OTM is inconsistent to their expection.

(2) Are all the baselines trained by the same loss?

(3) More neural operator baselines are expected.

(4) Demonstrate the novelty of STAC.

I think it is favorable that the authors experiment on extensive benchmarks. But I have some serious concerns. if the authors reply my questions properly, I am willing to raise my score.

---

> ### Author Response · Authors · 2023-11-19
> **Response to Reviewer rchA (I)**
>
> We are truly grateful for the time you have taken to review our paper and your insightful review. Here we address your comments in the following.
>
> > Q1. However, in my opinion, I think it is insufficient in novelty. STAC just combines a series of advanced models, including FNO, Vision Transformer, Neural ODE, and a similar recall mechanism proposed by E3D-LSTM. The proposed training strategy is also in a combination style. For me, it is hard to find the novel part in this model.
>
> A1. Thanks for your comment. The novelty of this methodology comes from three parts.
>
> -  **New complementary perspective**, which explores spatio-temporal dynamics in both discrete and continuous manners. They are both closely related to our target long-term predictions.
> -  **Information fusion strategy**, which includes fine-grained fusion and coarse-grained fusion to tackle feature maps with different granularities.
> - **Cache mechanism**, which aims to store previous states, thus allowing the system to **remember** and **reuse** historical data for future predictions. Compared with E3D-LSTM storing predictions at different timesteps, our cache mechanism **stores feature maps** with a long interval for long-term spatiotemporal predictions. Moreover, we involve more complicated interaction among current states, short-term states, and long-term states by **involving short-term interaction** map $A_m$ and updated maps $Q_m$. In contrast, E3D-LSTM utilizes a simple recurrent architecture, which achieves much worse performance.
>
> Besides, this work also the contribution of new dataset and new benchmark. In the future, we will fully open-source the data sets of all working conditions and provide the code for reading and processing.
>
> > Q2. Why should they combine vision transformer and FNO? FNet has shown that the feedforward layer can perform like FFT. Why not just only use Transformer or FNO?
>
> A2. Thanks for your comment. We have added four model variants as follows:
>
> - STAC w/o FNO, which removes FNO and only uses Transformer in our FSM module.
> - STAC w/o Transformer, which removes the Transformer module and only uses FNO in our FSM module.
> - STAC w/o FNO&Transformer +Fnet, which removes the FNO and Transformer modules and replaces them with Fnet.
> - Fnet, which only uses Fnet.
>
> The compared performance is shown below. From the results, we can find that both mechanisms have a crucial effect on performance. The reason is that Transformer focuses on capturing the correlation between pixels, while FNO learns function mapping in the frequency domain. Combining them can provide a comprehensive for semantics learning. In addition, incorporating Fnet into STAC and the predictions alone cannot achieve superior performance as well.
>
>
> | Model                          | MSE (Fire 10 - 10) | SSIM (Fire 10 - 10) | MSE/100 (KTH 10 - 20) | SSIM (KTH 10 -20) |
> | ------------------------------ | ------------------ | ------------------- | --------------------- | ----------------- |
> | STAC                           | 0.0487             | 92.87               | 0.2983                | 92.83             |
> | STAC w/o FNO                   | 0.0756             | 84.54               | 0.3021                | 90.79             |
> | STAC w/o Transformer           | 0.0506             | 91.21               | 0.4765                | 83.72             |
> | STAC w/o FNO&Transformer +Fnet | 0.1281             | 79.48               | 0.7325                | 78.23             |
> | Fnet                           | 0.7641             | 45.43               | 24.5432               | 56.41             |
>
> > Q3. Why can Neural ODE capture the continuous dynamics? I know that Neural ODE can achieve the adaptive depth or adaptive temporal interval. However, I think the usage here is equivalent to a simple rk4 algorithm. It is hard to claim that they learn the continuous dynamic feature. Besides, They don’t present the necessity in using Neural ODE.
>
> A3. Thanks for your comment. Our approach utilizes Neural ODE to explore spatiotemporal relationships in a continuous way, which can allow the capture of long-range correlations naturally without increasing the depth. This point is important for effective long-term predictions. We have added a model invariant STAC w/o OTM, which removes the ODE module. The performance is shown below. From the results, we can observe our ODE module is necessary for accurate long-term prediction.
>
> | Model        | MSE (Fire 10 - 10) | SSIM (Fire 10 - 10) | MSE/100 (KTH 10 - 20) | SSIM (KTH 10 - 20) | MSE x 1000 (SWE 20 - 140) |
> | ------------ | ------------------ | ------------------- | --------------------- | ------------------ | ------------------------- |
> | STAC w/o OTM | 0.0802             | 82.33               | 0.5644                | 78.43              | 2.153                     |
> | STAC         | 0.0487             | 92.87               | 0.2983                | 92.83              | 0.824                     |

---

> ### Author Response · Authors · 2023-11-19
> **Response to Reviewer rchA (II)**
>
> > Q4. Are the experimental datasets temporally irregular? According to the paper, I think the input sequences are equally collected along the temporal dimension.
>
> A4. Thanks for your comment. The dataset we use is temporally regular. The complete dataset statistics are as follows:
>
>   **Table: Dataset statistics.** *N_tr* and *N_te* denote the number of instances in the training and test sets. The lengths of the input and prediction sequences are *I_l* and *O_l*, respectively.
>
>   | **Dataset**     | **N_tr** | **N_te** | **(C, H, W)**  | **I_l** | **O_l** | **Interval** |
>   | --------------- | -------- | -------- | -------------- | ------- | ------- | ------------ |
>   | Turbulence      | 5000     | 1000     | (3, 300, 300)  | 50      | 50      | 1 second     |
>   | ERA5 (Global)   | 10000    | 2000     | (1, 1440, 720) | 10      | 10      | 1 day        |
>   | ERA5 (Local)    | 5000     | 1000     | (2, 200, 200)  | 8       | 8       | 1 hour       |
>   | KTH             | 108717   | 4086     | (1, 128, 128)  | 10      | 20      | 1 step       |
>   | SEVIR           | 4158     | 500      | (1, 384, 384)  | 13      | 12      | 5 mins       |
>   | Fire            | 6000     | 1500     | (2, 32, 480)   | 50      | 350      | 1 second     |
>   | SWE             | 2000     | 200      | (1, 128, 256)  | 20      | 140      | 1 second     |
>   | dynamics system | 6000     | 1200     | (3, 128, 128)  | 2       | 2       | 1 second     |
>
> > Q5. About the cache-based design. I think it is necessary to demonstrate its advancement over the temporal recall gate in E3D-LSTM.
>
> A5. Thanks for your comment. We have added a model variant STAC w/ E3D, which replaces our cache-based design with the temporal recall gate in E3D-LSTM. The compared results are shown below. From the results, we can observe that STAC performs the best in all cases. The reason is that our module not only stores feature maps with a long interval but also involves short-term interaction maps into spatio-temporal predictions.
>
> | Iuput-Output length                   | 50 - 400   | 50 - 400    | 20 - 140        | 20 - 140   | 10 - 40    | 10 - 40    |
> | ------------------------------------- | ---------- | ----------- | --------------- | ---------- | ---------- | ---------- |
> | Model                                 | MSE (Fire) | SSIM (Fire) | MSE x 100 (SWE) | SSIM (SWE) | PSNR (KTH) | SSIM (KTH) |
> | E3D-LSTM                              | 1.2983     | 75.8721     | 0.2298          | 76.3312    | 30.5931    | 87.4221    |
> | STAC                                  | 0.6631     | 88.8731     | 0.0824          | 97.6762    | 33.9831    | 92.1176    |
> | STAC w/ E3D | 0.9987     | 83.4730      | 0.1222          | 87.283     | 31.3984    | 88.3984    |
>
> > Q6. In addition to the performance, they should compare the efficiency with other baselines, including running time, GPU memory and parameter size.
>
> A6. Thanks for your comment. We have added the following table as follows. From the comparison, we can observe that our efficiency is competitive. The reason is that although we have a range of modules, the model depth is relatively small, which can save extensive computation costs.
>
> | Model      | Memory (MB) | FLOPs (G) | Params (M) | Training time |
> | ---------- | ----------- | --------- | ---------- | ------------- |
> | FNO        | 8.41        | 12.31     | 7.271      | 32s / epoch   |
> | F-FNO      | 12.3        | 13.12     | 11.21      | 76s / epoch   |
> | E3D-LSTM   | 2691        | 288.9     | 51         | 172s / epoch  |
> | MIM        | 2331        | 179.2     | 38         | 154s / epoch  |
> | PredRNN-V2 | 1721        | 117.3     | 23.9       | 126s / epoch  |
> | SimVP-V2   | 421         | 17.2      | 46.8       | 25s / epoch   |
> | LSM        | 10.21       | 14.31     | 9.002      | 37 s / epoch  |
> | U-NO       | 92          | 32.1      | 136        | 278 s / epoch |
> | STAC       | 578         | 22.81     | 25.4       | 98s / epoch   |
>
> >Q7. In the current version, they only compare STAC with video prediction baselines. How about the advanced neural operators, such as LSM [2], U-NO [3]?
>
> A7. Thanks for your comment. We have added LSM and U-NO for a more comprehensive evaluation. The experimental results show that the STAC model outperforms LSM and U-NO on different datasets, which validates the superiority of the proposed STAC.
>
> | Model      | STAC    | STAC    | LSM     | LSM     | U-NO    | U-NO    |
> | ---------- | ------- | ------- | ------- | ------- | ------- | ------- |
> | Dataset    | MSE     | MAE     | MSE     | MAE     | MSE     | MAE     |
> | Turbulence | 0.5123  | 0.5345  | 0.6412  | 0.7553  | 0.5654  | 0.6093  |
> | ERA5       | 1.9865  | 1.7791  | 3.9831  | 2.3132  | 3.4612  | 2.2931  |
> | SEVIR      | 1.9731  | 1.4054  | 2.9831  | 2.4431  | 2.2031  | 1.5632  |
> | Fire       | 0.5493  | 0.7217  | 1.2831  | 1.0932  | 1.5643  | 0.9853  |
> | KTH        | 28.8321 | 24.2216 | 39.9831 | 38.4432 | 35.8732 | 34.4322 |

---

> ### Author Response · Authors · 2023-11-19
> **Response to Reviewer rchA (III)**
>
> >Q7. In the current version, they only compare STAC with video prediction baselines. How about the advanced neural operators, such as LSM [2], U-NO [3]?
>
> A7. Thanks for your comment. We have added LSM and U-NO for a more comprehensive evaluation. The experimental results show that the STAC model outperforms LSM and U-NO on different datasets, which validates the superiority of the proposed STAC.
>
> | Model      | STAC    | STAC    | LSM     | LSM     | U-NO    | U-NO    |
> | ---------- | ------- | ------- | ------- | ------- | ------- | ------- |
> | Dataset    | MSE     | MAE     | MSE     | MAE     | MSE     | MAE     |
> | Turbulence | 0.5123  | 0.5345  | 0.6412  | 0.7553  | 0.5654  | 0.6093  |
> | ERA5       | 1.9865  | 1.7791  | 3.9831  | 2.3132  | 3.4612  | 2.2931  |
> | SEVIR      | 1.9731  | 1.4054  | 2.9831  | 2.4431  | 2.2031  | 1.5632  |
> | Fire       | 0.5493  | 0.7217  | 1.2831  | 1.0932  | 1.5643  | 0.9853  |
> | KTH        | 28.8321 | 24.2216 | 39.9831 | 38.4432 | 35.8732 | 34.4322 |
>
> >Q8. Are all the baselines trained by the same loss as STAC? This point is essential to ensure a fair comparison.
>
> A8. Thanks for your comment. The loss function of the baseline follows the original paper, which is MSE loss. Considering fairness, we train STAC with only MSE loss. The experimental results are shown in the table below. It can be seen that with only MSE loss, our STAC can outperform the baselines, and adding our designed loss would further increase the performance.
>
>
> | Dataset             | Fire   | Fire    | ERA5   | ERA5    |
> | ------------------- | ------ | ------- | ------ | ------- |
> | Model               | MSE    | SSIM    | MSE    | SSIM    |
> | STAC                | 0.5493 | 92.3192 | 1.9865 | 87.4531 |
> | STAC(only MSE Loss) | 0.6432 | 90.8752 | 2.1129 | 85.4853 |
> | PredRNN-V2          | 0.7789 | 85.9743 | 2.2731 | 84.9853 |
> | SimVP-V2            | 1.7743 | 77.8654 | 3.0843 | 77.3212 |
> | FNO                 | 0.9985 | 82.2218 | 2.8534 | 79.3834 |
> | LSM                 | 0.9654 | 83.9482 | 2.9831 | 79.4393 |

---

> ### Author Response · Authors · 2023-11-19
> **Response to Reviewer rchA (IV)**
>
> >Q9. More detailed ablations are expected. They should also conduct the following experiments:
>
> > Removing FNO or Transformer in FSM.
>
> > Replacing OTM with ConvLSTM or PredRNN.
>
> > Replacing the CRP with the recall gate in E3D-LSTM.
>
>
> A9. Thanks for your comment. We have added five model variants as follows:
>
> - STAC w/o FNO, which removes FNO and only uses Transformer in our FSM module.
> - STAC w/o Transformer, which removes the Transformer module and only uses FNO in our FSM module.
> - STAC (With ConvLSTM), which replaces the OTM module with ConvLSTM.
> - STAC (With PredRNN), which replaces the OTM module with PredRNN.
> - STAC (With Recall gate), which replaces the CRP with the Recall gate.
>
> Experiments show that removing FNO and Transformer from the STAC model reduces performance especially Transformer has a significant impact. Replacing OTM with ConvLSTM or PredRNN significantly degrades performance, especially PredRNN replacement leads to memory overflow. Replacing CRP with Recall gate also shows the importance of CRP, whose removal leads to performance loss.
>
>
>
> | Model                                 | MSE (Fire) | SSIM (Fire) | MSE/100 (KTH) | SSIM (KTH) |
> | ------------------------------------- | ---------- | ----------- | ------------- | ---------- |
> | STAC                                  | 0.0487     | 92.87       | 0.2983        | 92.83      |
> | STAC w/o FNO                          | 0.0756     | 84.54       | 0.3021        | 90.79      |
> | STAC w/o Transformer                  | 0.0506     | 91.21       | 0.4765        | 83.72      |
> | STAC (With ConvLSTM)    | 0.1723     | 65.69       | 1.2043        | 65.42      |
> | STAC (With PredRNN)     | 0.3922     | 57.98       | OOM           | OOM        |
> | STAC (With Recall gate) | 0.0998     | 79.49       | 0.5145        | 80.39      |
>
> In light of these responses, we hope we have addressed your concerns, and hope you will consider raising your score. If there are any additional notable points of concern that we have not yet addressed, please do not hesitate to share them, and we will promptly attend to those points.

---

> > ### Comment · Reviewer_rchA · 2023-11-21
> > **Thanks for your response**
> >
> > I appreciate the authors' effort in clarifying their novelty and providing sufficient ablations and baselines. They made a great effort obviously.
> >
> > Most of my concerns are resolved. However, the questions about continuous modeling still remain (Q3 and Q4).
> >
> > (1) I don't think the usage in STAC is Neural ODE. It is just a rk4 algorithm. If you do want to claim that your model takes benefits from Neural ODE, you need to use the learnable temporal steps $t$.
> >
> > (2) Would you please clarify this comment "which can allow the capture of long-range correlations naturally without increasing the depth" in your rebuttal? What kind of long-range correlations do you mean?
> >
> > Overall, I think the ODE claim in this paper is not rigorous, which obviously requires a huge revision to the paper. Thus, I would like to raise my score to 5 but cannot give an acceptance recommendation.

---

> ### Author Response · Authors · 2023-11-22
> **Thanks for your feedback!**
>
> Thanks for your feedback and we are happy to resolve your further concerns as follows:
>
> > Q1. I don't think the usage in STAC is Neural ODE. It is just a rk4 algorithm. If you do want to claim that your model takes benefits from Neural ODE, you need to use the learnable temporal steps $t$.
>
> A1. Thanks for your comment. Clearly, the earliest neural ODE [1] also focuses on the fixed depth, but introduces the continuous propagation with benefits. Moreover, [2] also utilizes the fixed step $t$ with performance improvement, which shows the benefit of adopting neural ODE with fixed steps. Moreover, we would include learnable $t$ in our future work.
>
> > Would you please clarify this comment "which can allow the capture of long-range correlations naturally without increasing the depth" in your rebuttal? What kind of long-range correlations do you mean?
>
> A2. Thanks for your comment. Here, the long-range correlations mean the relationships of pixels far away in tensor $\bar{I}^{in}$, which require a large receptive field to capture. Here, neural ODE utilizes continuous propagation to get rid of stacking a range of CNN blocks.
>
> Thanks again for appreciating our work and for your constructive suggestions. Please let us know if you have further questions.
>
> [1] Chen et al. Neural Ordinary Differential Equations.
>
> [2] Jin et al., Multivariate Time Series Forecasting with Dynamic Graph Neural ODEs

---

> > ### Author Response · Authors · 2023-11-22
> > **UPDATED! Thanks for your feedback and we tried learnable t!**
> >
> > Thanks for your feedback and we are happy to resolve your further concerns as follows:
> >
> > > Q1. I don't think the usage in STAC is Neural ODE. It is just a rk4 algorithm. If you do want to claim that your model takes benefits from Neural ODE, you need to use the learnable temporal steps $t$.
> >
> > A1. Thanks for your comment. Here, **we have tried the learnable $t$ with compared performance as follows**. The results show that the learnable $t$ can slightly improve the performance compared to the fixed $t$ with more flexibility. Thanks for your suggestion. We have added the results into our manuscript.  (A smaller mse means better model performance and a larger ssim means better model performance.)
> >
> > Besides, the earliest neural ODE [1] also focuses on the fixed depth, but introduces the continuous propagation with benefits. Moreover, [2] also utilizes the fixed step $t$ with performance improvement, which shows the benefit of adopting neural ODE in both settings.
> >
> > | Model                | MSE (Fire 10 - 10) | SSIM (Fire 10 - 10) | MSE/100 (KTH 10 - 20) | SSIM (KTH 10 - 20) | MSE x 1000 (SWE 20 - 140) | SSIM x 1000(SWE 20 - 140) | Avg Training Time |
> > | -------------------- | ------------------ | ------------------- | --------------------- | ------------------ | ------------------------- | ------------------------- | ----------------- |
> > | STAC                 | 0.0487             | 92.87               | 0.2983                | 92.83              | 0.824                     | 97.68                     | 242s / epoch      |
> > | STAC (learnable $t$) | 0.0464             | 93.02               | 0.2873                | 93.11              | 0.814                     | 98.53                     | 257s / epoch       |
> >
> >
> >
> > > Q2. Would you please clarify this comment "which can allow the capture of long-range correlations naturally without increasing the depth" in your rebuttal? What kind of long-range correlations do you mean?
> >
> > A2. Thanks for your comment. Here, the long-range correlations mean the relationships of pixels far away in tensor $\bar{I}^{in}$, which require a large receptive field to capture. Here, neural ODE utilizes continuous propagation to get rid of stacking a range of CNN blocks.
> >
> > Thanks again for appreciating our work and for your constructive suggestions. Please let us know if you have further questions.
> >
> > [1] Chen et al. Neural Ordinary Differential Equations.
> >
> > [2] Jin et al., Multivariate Time Series Forecasting with Dynamic Graph Neural ODEs

---

> ### Author Response · Authors · 2023-11-23
> **Looking forward to your reply !**
>
> Dear Reviewer rchA,
>
> Thank you very much again for the time and effort put into reviewing our paper. We believe that we have addressed all your concerns in our response. We have also followed your suggestion to improve our paper and have added additional experimental analysis. We kindly remind you that we are approaching the end of the discussion period. We would love to know if there is any further concern, additional experiments, suggestions, or feedback,  we kindly hope that you can consider increasing the score.
>
> Best regards,
>
> All authors

---

> > ### Author Response · Authors · 2023-11-23
> > **UPDATED! Looking forward to your reply.**
> >
> > Dear reviewers,
> >
> > We sincerely appreciate your valuable feedback. As the deadline for the author-reviewer discussion phase is approaching, we would like to check if our response adequately addressed your concerns regarding our neural ODE and learnable $t$. If there are any remaining issues or if you require further clarification, please feel free to inform us.
> >
> > Best Regards,
> >
> > Authors.

---

> > > ### Comment · Reviewer_rchA · 2023-11-23
> > > **Thanks for your response**
> > >
> > > Dear reviewer,
> > >
> > > Thanks for your response and try the learnable t. But for me, the description of "which can allow the capture of long-range correlations naturally without increasing the depth" is still not well-supported. Even though you can "increase" the depth in favorable efficiency with NODE, the reception field may not increase as you want.
> > >
> > > Considering the novelty and soundness of this paper, I would like to keep my original score.

---

> ### Author Response · Authors · 2023-11-23
> **Thank you for your invaluable feedback!**
>
> Thanks for your feedback and we are happy to resolve your further concerns as follows:
>
> > Q1. The description of "which can allow the capture of long-range correlations naturally without increasing the depth" is still not well-supported.
>
> A1. Thanks for your comment. Here, we revised our statement into "which can allow the capture of long-range correlations naturally with robustness" since Neural ODE has the property of robustness with increased propagation depths, as stated in Sec. 5.4 of [1]. This is one of the most important properties of Neural ODE compared with discrete methods. We have revised the manuscript accordingly.
>
> [1] Multivariate Time Series Forecasting with Dynamic Graph Neural ODEs
>
> > Q2. About the soundness.
>
> A2. Thanks for your comment. We have added extensive clarification and experiments to address all the concerns. Moreover, the other reviewers all give a "3" for soundness. If there are any remaining issues or if you require further clarification, please feel free to inform us.
>
> > Q3. About the novelty.
>
> A3. Thanks for your comment. As mentioned earlier, our method consists of three parts, i.e., complementary perspective, information fusion strategy and cache mechanism, which are novel compared with existing works. This work also the contribution of **new dataset and new benchmark in advancing the community**. And we are committed to **making our complete code and the dataset publicly available.**
>
> As the deadline for the author-reviewer discussion phase is approaching, we hope to make sure that our response sufficiently addressed your concerns regarding the revised version. We hope this could align with your expectations and positively influence the score. Please do not hesitate to let us know if you need any clarification or have additional suggestions.

---

### Author Response · Authors · 2023-11-19
**Thank all reviewers for your careful reviews and constructive suggestions!**

Dear Reviewers,

We thank you for your careful reviews and constructive suggestions. We acknowledge the positive comments such as "**technologically reasonable**” (Reviewer rchA), "**clearly presented and well-written**" (Reviewer rchA), "**a wide range of benchmarks**" (Reviewer rchA), "**an important research problem**,” (Reviewer CD63), "**a well-prepared benchmark**” (Reviewer CD63), "**extensive experiments**” (Reviewer CD63), "**effective modeling**” (Reviewer zrZ8), “**Comprehensive experimental validation**” (Reviewer zrZ8), “**Effective optimization strategies**” (Reviewer zrZ8), "**novel perspective**" (Reviewer UcXH), "**intuitive and reasonable**" (Reviewer UcXH), "**well organized and clearly written**" (Reviewer UcXH), "**new fire dynamics benchmark**" (Reviewer UcXH) and "**comprehensive experiments**" (Reviewer UcXH). We also believe that our **new benchmark FIRE** would contribute to the community more.

We have made extensive revisions based on these valuable comments, which we briefly summarize below for your convenience, and have accordingly revised our article with major changes highlighted in blue.

- We have clarified our novelty in four points including **new benchmark FIRE**, **new complementary perepsectives for long-term prediction**, **information fusion strategy** and **new cache mechanism with differences stated**.

- We have introduced detailed motivation of each component related to real data to show these **compotents typically focus on long-term dependency mining** for our long-term prediction tasks.

- We have added more **sufficient ablation studies** to show the effectiveness and reliability of each component.

- We have included more **competing baselines** including LSM and U-NO to demonstrate the superiority of our approach.


- We have included more detailed **efficiency comparisons** between our method and various baselines and observed that our method shows competitive efficiency.

- We have **proofread** the manuscript to correct some typos and mistakes.

We have also responded to your questions point by point. Once again, we greatly appreciate your effort and time spent revising our manuscript. In summary, our paper introduces **a new benchmark dataset**, extensive performance comparison with **14 benchmarks**, and a **practical approach** to achieve superior long-term predictive performance, which we believe can greatly contribute to the community. Please let us know if you have any follow-up questions. We will be happy to answer them.

Best regards,

the Authors

---

### Comment · Area_Chair_v5sU · 2023-11-23
**From AC at the end of rebuttal: Reviewer response required**

Dear Reviewers,

Thanks for your time and commitment to the ICLR 2024 review process.

As we approach the conclusion of the author-reviewer discussion period (Wednesday, Nov 22nd, AOE), I kindly urge those who haven't engaged with the authors' dedicated rebuttal to please take a moment to review their response and share your feedback, regardless of whether it alters your opinion of the paper.

Your feedback is essential to a thorough assessment of the submission.

Best regards,

AC

---

### Meta-Review · Area_Chair_v5sU · 2023-12-10

**Metareview:**

This paper proposes a composite solution to modeling long-term dynamics, which combines Vision Transformer, NODE, and Cache mechanism, and more. Authors also provided lengthy rebuttal with valuable experimental results, which make the paper strong in the empirical aspect, and were acknowledged by reviewers with increased scores. After rebuttal, reviewers still held crucial concerns towards this paper. Most reviewers felt that the novelty is relatively slim, given that most of the components are inherited from previous works. The introduction of NODE and its actual role in the final solution is not clear. The cache design is incremental compared with previous works. The overall solution is way more complicated than the previous counterparts. In a nutshell, the paper needs to undergo a major revision.

**Justification For Why Not Higher Score:**

After rebuttal, unanimous rejection. Novelty is not pronounced enough.

**Justification For Why Not Lower Score:**

N/A

---

### Decision · Program_Chairs · 2024-01-16

Reject